# EXPLAINING TIME SERIES VIA CONTRASTIVE AND LOCALLY SPARSE PERTURBATIONS

**Zichuan Liu**[1,2*]**, Yingying Zhang**[2*]**, Tianchun Wang**[3*]**, Zefan Wang**[2,4]**, Dongsheng Luo**[5]**,
Mengnan Du**[6]**, Min Wu**[7]**, Yi Wang**[8]**, Chunlin Chen**[1†]**, Lunting Fan**[2]**, and Qingsong Wen**[2†]
[1]Nanjing University, [2]Ailibaba Group, [3]Pennsylvania State University,
[4]Tsinghua University, [5]Florida International University,
[6]New Jersey Institute of Technology, [7]A*STAR, [8]The University of Hong Kong

## ABSTRACT

Explaining multivariate time series is a compound challenge, as it requires identifying important locations in the time series and matching complex temporal patterns. Although previous saliency-based methods addressed the challenges, their perturbation may not alleviate the distribution shift issue, which is inevitable especially in heterogeneous samples. We present ContraLSP, a locally sparse model that introduces counterfactual samples to build uninformative perturbations but keeps distribution using contrastive learning. Furthermore, we incorporate sample-specific sparse gates to generate more binary-skewed and smooth masks, which easily integrate temporal trends and select the salient features parsimoniously. Empirical studies on both synthetic and real-world datasets show that ContraLSP outperforms state-of-the-art models, demonstrating a substantial improvement in explanation quality for time series data. The source code is available at `https://github.com/zichuan-liu/ContraLSP`.

## 1 INTRODUCTION

Providing reliable explanations for predictions made by machine learning models is of paramount importance, particularly in fields like finance (Mokhtari et al., 2019), games (Liu et al., 2023), and healthcare (Amann et al., 2020), where transparency and interpretability are often ethical and legal prerequisites. These domains frequently deal with complex multivariate time series data, yet the investigation into methods for explaining time series models remains an underexplored frontier (Rojat et al., 2021). Besides, adapting explainers originally designed for different data types presents challenges, as their inductive biases may struggle to accommodate the inherently complex and less interpretable nature of time series data (Ismail et al., 2020). Achieving this requires the identification of crucial temporal positions and aligning them with explainable patterns.

In response, the predominant explanations involve the use of saliency methods (Baehrens et al., 2010; Tjoa & Guan, 2020), where the explanatory distinctions depend on how they interact with an arbitrary model. Some works establish saliency maps, e.g., incorporating gradient (Sundararajan et al., 2017; Lundberg et al., 2018) or constructing attention (Garnot et al., 2020; Lin et al., 2020), to better handle time series characteristics. Other surrogate methods, including Shapley (Castro et al., 2009; Lundberg & Lee, 2017) or LIME (Ribeiro et al., 2016), provide insight into the predictions of a model by locally approximating them through weighted linear regression. These methods mainly provide instance-level saliency maps, but the feature inter-correlation often leads to notable generalization errors (Yang et al., 2022).

The most popular class of explanation methods is to use samples for perturbation (Fong et al., 2019; Leung et al., 2023; Lee et al., 2022), usually through different styles to make non-salient features uninformative. Two representative perturbation methods in time series are Dynamask (Crabbé & Van Der Schaar, 2021) and Extrmask (Enguehard, 2023).

---

\* Authors contributed equally.
† Correspondence to `qingsongedu@gmail.com` and `clchen@nju.edu.cn`.

Dynamask utilizes meaningful perturbations to incorporate temporal smoothing, while Extrmask generates perturbations of less sense close to zero through neural network learning. However, due to shifts in shape (Zhao et al., 2022), perturbed time series may be out of distribution for the explained model, leading to a loss of faithfulness in the generated explanations. For example, a time series classified as 1 and its different forms of perturbation are shown in Figure 1. We see that the distribution of all classes moves away from 0 at intermediate time, while the 0 and mean perturbations shift in shape. In addition, the blur and learned perturbations are close to the original feature and therefore contain information for classification 1. It may result in a label leaking problem (Jethani et al., 2023), as informative perturbations are introduced. This causes us to think about counterfactuals, i.e., a contrasting perturbation does not affect model inference in non-salient areas.

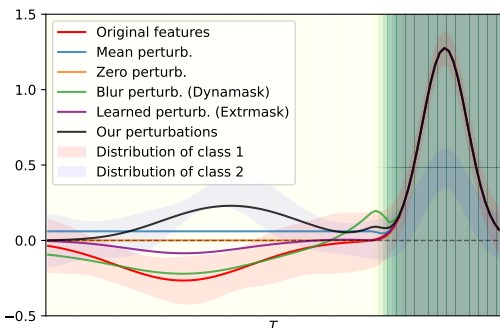

Figure 1: Illustrating different styles of perturbation. The red line is a sample belonging to class 1 within the two categories, while the dark background indicates the salient features, otherwise non-salient. Other perturbations could be either not uninformative or not in-domain, while ours is counterfactual that is toward the distribution of negative samples.

To address these challenges, we propose a Contrastive and Locally Sparse Perturbations (ContraLSP) framework based on contrastive learning and sparse gate techniques. Specifically, our ContraLSP learns a perturbation function to generate counterfactual and in-domain perturbations through a contrastive learning loss. These perturbations tend to align with negative samples that are far from the current features (Figure 1), rendering them uninformative. To optimize the mask, we employ $\ell_0$-regularised gates with injected random noises in each sample for regularization, which encourages the mask to approach a binary-skewed form while preserving the localized sparse explanation. Additionally, we introduce a smooth constraint with a trend function to allow the mask to capture temporal patterns. We summarize our contributions as below:

- We propose ContraLSP as a stronger time series model explanatory tool, which incorporates counterfactual samples to build uninformative in-domain perturbation through contrastive learning.

- ContraLSP integrates sample-specific sparse gates as a mask indicator, generating binary-skewed masks on time series. Additionally, we enforce a smooth constraint by considering temporal trends, ensuring consistent alignment of the latent time series patterns.

- We evaluate our method through experiments on both synthetic and real-world time series datasets. These datasets comprise either classification or regression tasks and the synthetic one includes ground-truth explanatory labels, allowing for quantitative benchmarking against other state-of-the-art time series explainers.

## 2 RELATED WORK

**Time series explainability.** Recent literature has delved into the realm of eXplainable Artificial Intelligence (XAI) for multivariate time series (Bento et al., 2021; Ismail et al., 2020; Zhu et al., 2023). Among them, gradient-based methods (Shrikumar et al., 2017; Sundararajan et al., 2017; Lundberg et al., 2018) translate the impact of localized input alterations to feature saliency. Attention-based methods (Lin et al., 2020; Choi et al., 2016) leverage attention layers to produce importance scores that are intrinsically based on attention coefficients. Perturbation-based methods, as the most common form in time series, usually modify the data through baseline (Suresh et al., 2017), generative models (Tonekaboni et al., 2020; Leung et al., 2023), or making the data more uninformative (Crabbé & Van Der Schaar, 2021; Enguehard, 2023). However, these methods provide only an instance-level saliency map, while the inter-sample saliency maps have been studied little in the existing literature (Gautam et al., 2022). Our investigation performs counterfactual perturbations through inter-sample variation, which goes beyond the instance-level saliency methods by focusing on understanding both the overall and specific model's behavior across groups.

**Model sparsification.** For a better understanding of which part of the features are most influential to the model's behavior, the existing literature enforcing sparsity (Fong et al., 2019) to constrain the model's focus on specific regions. A typical approach is LASSO (Tibshirani, 1996), which selects a subset of the most relevant features by adding an $\ell_1$ constraint to the loss function. Based on this, several works (Feng & Simon, 2017; Scardapane et al., 2017; Louizos et al., 2018; Yamada et al., 2020) are proposed to employ distinct forms of regularization to encourage the input features to be sparse. All these methods select global informative features that may neglect the underlying correlation between them. To cope with it, local stochastic gates (Yang et al., 2022) consider an instance-wise selector to heterogeneous samples, accommodating cases where salient features vary among samples. Lee et al. (2022) takes a self-supervised way to enhance stochastic gates that encourage the model's sparse explainability meanwhile. However, most of these sparse methods are utilized in tabular feature selection. Different from them, our approach reveals crucial features within the temporal patterns of multivariate time series data, offering local sample explanations.

**Counterfactual explanations.** Perturbation-based methods are known to have distribution shift problems, leading to abnormal model behaviors and unreliable explanation (Hase et al., 2021; Hsieh et al., 2021). Previous works (Goyal et al., 2019; Teney et al., 2020) have tackled generating reasonable counterfactuals for perturbation-based explanations, which searches pairwise inter-class perturbations in the sample domain to explain the classification models. In the field of time series, Delaney et al. (2021) builds counterfactuals by adapting label-changing neighbors. To alleviate the need for labels in model interpretation, Chuang et al. (2023) uses triplet contrastive representation learning with disturbed samples to train an explanatory encoder. However, none of these methods explored label-free perturbation generation aligned with the sample domain. On the contrary, our method yields counterfactuals with contrastive sample selection to sustain faithful explanations.

## 3 PROBLEM FORMULATION

Let $\{(\boldsymbol{x}_i, y_i)\}_{i=1}^N$ be a set of multi-variate time series, where $\boldsymbol{x}_i \in \mathbb{R}^{T \times D}$ is a sample with $T$ time steps and $D$ observations, $y_i \in \mathcal{Y}$ is the ground truth. $\boldsymbol{x}_i[t, d]$ denotes a feature of $\boldsymbol{x}_i$ in time step $t$ and observation dimension $d$, where $t \in [1 : T]$ and $d \in [1 : D]$. We let $\boldsymbol{x} \in \mathbb{R}^{N \times T \times D}, y \in \mathcal{Y}^N$ be the set of all the samples and that of the ground truth, respectively. We are interested in explaining the prediction $\hat{y} = f(\boldsymbol{x})$ of a pre-trained black-box model $f$. More specifically, our objective is to pinpoint a subset $\mathcal{S} \subseteq [N \times T \times D]$, in which the model uses the relevant selected features $\boldsymbol{x}[\mathcal{S}]$ to optimize its proximity to the target outcome. It can be rewritten as addressing an optimization problem: $\arg\min_{\mathcal{S}} \mathcal{L}(\hat{y}, f(\boldsymbol{x}[\mathcal{S}]))$, where $\mathcal{L}$ represents the cross-entropy loss for $C$-classification tasks (i.e., $\mathcal{Y} = \{1, \ldots, C\}$) or the mean squared error for regression tasks (i.e., $\mathcal{Y} = \mathbb{R}$).

To achieve this goal, we consider finding masks $\boldsymbol{m} = \mathbf{1}_{\mathcal{S}} \in \{0, 1\}^{N \times T \times D}$ by learning the samples of perturbed features through $\Phi(\boldsymbol{x}, \boldsymbol{m}) = \boldsymbol{m} \odot \boldsymbol{x} + (\mathbf{1} - \boldsymbol{m}) \odot \boldsymbol{x}^r$, where $\boldsymbol{x}^r = \varphi_{\theta_1}(\boldsymbol{x})$ is the counterfactual explanation obtained from a perturbation function $\varphi_{\theta_1} : \mathbb{R}^{N \times T \times D} \rightarrow \mathbb{R}^{N \times T \times D}$, and $\theta_1$ is a parameter of the function $\varphi(\cdot)$ (e.g., neural networks). Thus, existing literature (Fong & Vedaldi, 2017; Fong et al., 2019; Crabbé & Van Der Schaar, 2021) propose to rewrite the above optimization problem by learning an optimal mask as

$$\arg\min_{\boldsymbol{m}, \theta_1} \mathcal{L}(f(\boldsymbol{x}), f \circ \Phi(\boldsymbol{x}, \boldsymbol{m})) + \mathcal{R}(\boldsymbol{m}) + \mathcal{A}(\boldsymbol{m}), \tag{1}$$

which promotes proximity between the predictions on the perturbed samples and the original ones in the first term, and restricts the number of explanatory features in the second term (e.g., $\mathcal{R}(\boldsymbol{m}) = \|\boldsymbol{m}\|_1$). The third term enforces the mask's value to be smooth by penalizing irregular shapes.

**Challenges.** In the real world, particularly within the healthcare field, two primary challenges are encountered: (i) Current strategies (Fong & Vedaldi, 2017; Louizos et al., 2018; Lee et al., 2022; Enguehard, 2023) of learning the perturbation $\varphi(\cdot)$ could be either not counterfactual or out of distributions due to unknown data distribution (Jethani et al., 2023). (ii) Under-considering the inter-correlation of samples would result in significant generalization errors (Yang et al., 2022). During training, cross-sample interference among masks $\{\boldsymbol{m}_i\}_{i=1}^N$ may cause ambiguous sample-specific predictions, while local sparse weights can remove the ambiguity (Yamada et al., 2017). These challenges motivate us to learn counterfactual perturbations that are adapted to each sample individually with localized sparse masks.

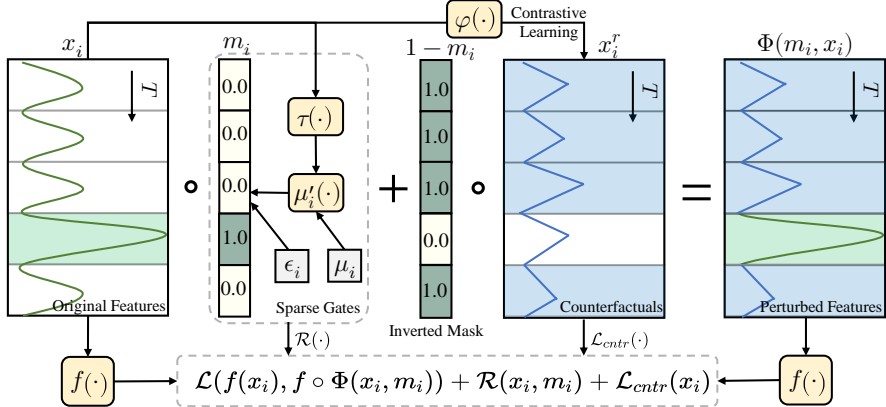

Figure 2: The architecture of ContraLSP. A sample of features $\boldsymbol{x}_i \in \mathbb{R}^{T \times D}$ is fed simultaneously to a perturbation function $\varphi(\cdot)$ and to a trend function $\tau(\cdot)$. The perturbation function $\varphi(\cdot)$ uses $\boldsymbol{x}_i$ to generate counterfactuals $\boldsymbol{x}_i^r$ that are closer to other negative samples (but within the sample domain) through contrastive learning. In addition, $\tau(\cdot)$ learns to predict temporal trends, which together with a set of parameters $\boldsymbol{\mu}_i$ depicts the smooth vectors $\boldsymbol{\mu}_i'$. It acts on the locally sparse gates by injecting noises $\boldsymbol{\epsilon}_i$ to get the mask $\boldsymbol{m}_i$. Finally, the counterfactuals are replaced with perturbed features and the predictions are compared to the original results to determine which features are salient enough.

## 4 OUR METHOD

We now present the Contrastive and Locally Sparse Perturbations (ContraLSP), whose overall architecture is illustrated in Figure 2. Specifically, our ContraLSP learns counterfactuals by means of contrastive learning to augment the uninformative perturbations but maintain sample distribution. This allows perturbed features toward a negative distribution in heterogeneous samples, thus increasing the impact of the perturbation. Meanwhile, a mask selects sample-specific features in sparse gates, which is learned to be constrained with $\ell_0$-regularization and temporal trend smoothing. Finally, comparing the perturbed prediction to the original prediction, we subsequently backpropagate the error to learn the perturbation function and adapt the saliency scores contained in the mask.

### 4.1 COUNTERFACTUALS FROM CONTRASTIVE LEARNING

To obtain counterfactual perturbations, we train the perturbation function $\varphi_{\theta_1}(\cdot)$ through a triplet-based contrastive learning objective. The main idea is to make counterfactual perturbations more uninformative by inversely optimizing a triplet loss (Schroff et al., 2015), which adapts the samples by replacing the masked unimportant regions. Specifically, we take each counterfactual perturbation $\boldsymbol{x}_i^r = \varphi_{\theta_1}(\boldsymbol{x}_i)$ as an anchor, and partition all samples $\boldsymbol{x}^r$ into two clusters: a positive cluster $\Omega^+$ and negative one $\Omega^-$, based on the pairwise Manhattan similarities between these perturbations. Following this partitioning, we select the $K^+$ nearest positive samples from the positive cluster $\Omega^+$, denoted as $\{\boldsymbol{x}_{i,k}^{r^+}\}_{k=1}^{K^+}$, which ex-

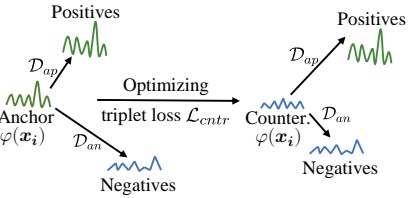

Figure 3: Illustration of the impact of triplet loss to generate counterfactual perturbations. The anchor is closer to negatives but farther from positives.

hibits similarity with the anchor features. In parallel, we randomly select $K^-$ subsamples from the negative cluster $\Omega^-$, denoted as $\{\boldsymbol{x}_{i,k}^{r^-}\}_{k=1}^{K^-}$, where $K^+$ and $K^-$ represent the numbers of positive and negative samples selected, respectively. The strategy of triple sampling is similar to Li et al. (2021), and we introduce the details in Appendix B.

To this end, we obtain the set of triplets $\mathcal{T}$ with each element being a tuple $\mathcal{T}_i = \left(\boldsymbol{x}_i^r, \{\boldsymbol{x}_{i,k}^{r^+}\}_{k=1}^{K^+}, \{\boldsymbol{x}_{i,k}^{r^-}\}_{k=1}^{K^-}\right)$. Let the Manhattan distance between the anchor with negative samples be $\mathcal{D}_{an} = \frac{1}{K^-}\sum_{k=1}^{K^-}|\boldsymbol{x}_i^r - \boldsymbol{x}_{i,k}^{r^-}|$, and that with positive samples be $\mathcal{D}_{ap} = \frac{1}{K^+}\sum_{k=1}^{K^+}|\boldsymbol{x}_i^r - \boldsymbol{x}_{i,k}^{r^+}|$.

As shown in Figure 3, we aim to ensure that $\mathcal{D}_{an}$ is smaller than $\mathcal{D}_{ap}$ with a margin $b$, thus making the perturbations counterfactual. Therefore, the objective of optimizing the perturbation function $\varphi_{\theta_1}(\cdot)$ with triplet-based contrastive learning is given by

$$\mathcal{L}_{cntr}(\boldsymbol{x}_i) = \max(0, \mathcal{D}_{an} - \mathcal{D}_{ap} - b) + \|\boldsymbol{x}_i^r\|_1, \tag{2}$$

which encourages the original sample $\boldsymbol{x}_i$ and the perturbation $\boldsymbol{x}_i^r$ to be dissimilar. The second regularization limits the extent of counterfactuals. In practice, the margin $b$ is set to 1 following (Balntas et al., 2016), and we discuss the effects of different distances in Appendix E.1.

## 4.2 SPARSE GATES WITH SMOOTH CONSTRAINT

Logical masks preserve the sparsity of feature selection but introduce a large degree of variance in the approximated Bernoulli masks due to their heavy-tailedness (Yamada et al., 2020). To address this limitation, we apply a sparse stochastic gate to each feature in each sample $i$, thus approximating the Bernoulli distribution for the local sample. Specifically, for each feature $\boldsymbol{x}_i[t, d]$, a sample-specific mask is obtained based on the hard thresholding function by

$$\boldsymbol{m}_i[t, d] = \min\left(1, \max(0, \boldsymbol{\mu}_i'[t, d] + \boldsymbol{\epsilon}_i[t, d])\right), \tag{3}$$

where $\boldsymbol{\epsilon}_i[t, d] \sim \mathcal{N}(0, \delta^2)$ is a random noise injected into each feature. We fix the Gaussian variance $\delta^2$ during training. Typically, $\boldsymbol{\mu}_i'[t, d]$ is taken as an intrinsic parameter of the sparse gate. However, as a binary-skewed parameter, $\boldsymbol{\mu}_i'[t, d]$ does not take into account the smoothness, which may lose the underlying trend in temporal patterns. Inspired by Elfwing et al. (2018) and Biswas et al. (2022), we adopt a sigmoid-weighted unit with the temporal trend to smooth $\boldsymbol{\mu}_i'$. Specifically, we construct the smooth vectors $\boldsymbol{\mu}_i'$ as

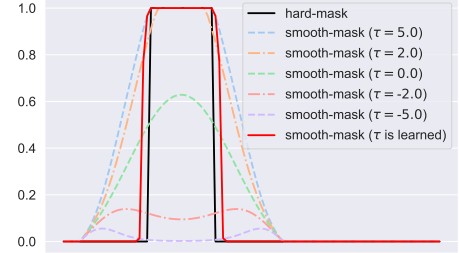

Figure 4: Different temperatures for the sigmoid-weighted unit. The learned trend function $\tau(\cdot)$ can be better adapted to smooth vectors (red) to hard masks (black).

$$\boldsymbol{\mu}_i' = \boldsymbol{\mu}_i \odot \sigma(\tau_{\theta_2}(\boldsymbol{x}_i)\boldsymbol{\mu}_i) = \frac{\boldsymbol{\mu}_i}{1 + e^{-\tau_{\theta_2}(\boldsymbol{x}_i)\boldsymbol{\mu}_i}}, \tag{4}$$

where $\tau_{\theta_2}(\cdot) : \mathbb{R}^{N \times T \times D} \to \mathbb{R}^{N \times T \times D}$ is a trend function parameterized by $\theta_2$ that plays a role in the sigmoid function as temperature scaling, and $\boldsymbol{\mu}_i$ is a set of parameters initialized randomly. In practice, we use a neural network (e.g., MLP) to implement the trend function $\tau_{\theta_2}(\cdot)$, whose details are shown in Appendix D.4. Note that employing a constant temperature may render the mask continuous. However, for a valid mask interpretation, adherence to a discrete property is appropriate (Queen et al., 2023). We illustrate in Figure 4 that a learned temperature (red solid) makes the hard mask smoother and keeps its skewed binary, in contrast to other constant temperatures.

To make the mask more informative in Eq. (1), we follow Yang et al. (2022) by replacing the $\ell_1$-regularization into an $\ell_0$-like constraint. Consequently, the regularization term $\mathcal{R}(\cdot)$ can be rewritten using the Gaussian error function (erf) as

$$\mathcal{R}(\boldsymbol{x}_i, \boldsymbol{m}_i) = \|\boldsymbol{m_i}\|_0 = \sum_{t=1}^{T} \sum_{d=1}^{D} \left(\frac{1}{2} + \frac{1}{2} \operatorname{erf}\left(\frac{\boldsymbol{\mu}_i'[t, d]}{\sqrt{2}\delta}\right)\right), \tag{5}$$

where $\boldsymbol{\mu}_i'$ is obtained from Eq.( 4). The full derivations are given in Appendix A. We calculate the empirical expectation over $\boldsymbol{m}_i$ for all samples. Thus, masks $\boldsymbol{m}$ are learned by the objective

$$\arg\min_{\boldsymbol{\mu}, \theta_2} \mathcal{L}\left(f(\boldsymbol{x}), f \circ \Phi(\boldsymbol{x}, \boldsymbol{m})\right) + \frac{\alpha}{N} \sum_{i=1}^{N} \mathcal{R}(\boldsymbol{x}_i, \boldsymbol{m}_i), \tag{6}$$

where $\alpha$ is the regular strength. Note that the smooth vectors $\boldsymbol{\mu}_i'$ restrict the penalty term $\mathcal{A}(\cdot)$ in Eq. (1) for jump saliency over time.

Table 1: Performance on Rare-Time and Rare-Observation experiments w/o different groups.

| METHOD | RARE-TIME | | | | RARE-TIME (DIFFGROUPS) | | | |
|---|---|---|---|---|---|---|---|---|
| | AUP↑ | AUR↑ | $I_m/10^4$↑ | $S_m/10^2$↓ | AUP↑ | AUR↑ | $I_m/10^4$↑ | $S_m/10^2$↓ |
| FO | $\mathbf{1.00}_{\pm 0.00}$ | $0.13_{\pm 0.00}$ | $0.46_{\pm 0.01}$ | $47.20_{\pm 0.61}$ | $\mathbf{1.00}_{\pm 0.00}$ | $0.16_{\pm 0.00}$ | $0.53_{\pm 0.01}$ | $54.89_{\pm 0.70}$ |
| AFO | $\mathbf{1.00}_{\pm 0.00}$ | $0.15_{\pm 0.01}$ | $0.51_{\pm 0.01}$ | $55.60_{\pm 0.85}$ | $\mathbf{1.00}_{\pm 0.00}$ | $0.16_{\pm 0.00}$ | $0.54_{\pm 0.01}$ | $57.76_{\pm 0.72}$ |
| IG | $\mathbf{1.00}_{\pm 0.00}$ | $0.13_{\pm 0.00}$ | $0.46_{\pm 0.01}$ | $47.61_{\pm 0.62}$ | $\mathbf{1.00}_{\pm 0.00}$ | $0.15_{\pm 0.00}$ | $0.53_{\pm 0.01}$ | $54.62_{\pm 0.85}$ |
| SVS | $\mathbf{1.00}_{\pm 0.00}$ | $0.13_{\pm 0.00}$ | $0.47_{\pm 0.01}$ | $47.20_{\pm 0.61}$ | $\mathbf{1.00}_{\pm 0.00}$ | $0.15_{\pm 0.00}$ | $0.52_{\pm 0.02}$ | $54.28_{\pm 0.84}$ |
| DYNAMASK | $\underline{0.99}_{\pm 0.01}$ | $0.67_{\pm 0.02}$ | $8.68_{\pm 0.11}$ | $37.24_{\pm 0.48}$ | $\underline{0.99}_{\pm 0.01}$ | $0.51_{\pm 0.00}$ | $5.75_{\pm 0.13}$ | $47.33_{\pm 1.02}$ |
| EXTRMASK | $\mathbf{1.00}_{\pm 0.00}$ | $\underline{0.88}_{\pm 0.00}$ | $\underline{16.40}_{\pm 0.13}$ | $\underline{13.10}_{\pm 0.78}$ | $\mathbf{1.00}_{\pm 0.00}$ | $\underline{0.83}_{\pm 0.03}$ | $\underline{13.37}_{\pm 0.78}$ | $\underline{27.44}_{\pm 3.68}$ |
| CONTRALSP | $\mathbf{1.00}_{\pm 0.00}$ | $\mathbf{0.97}_{\pm 0.01}$ | $\mathbf{19.51}_{\pm 0.30}$ | $\mathbf{4.65}_{\pm 0.71}$ | $\mathbf{1.00}_{\pm 0.00}$ | $\mathbf{0.94}_{\pm 0.01}$ | $\mathbf{18.92}_{\pm 0.37}$ | $\mathbf{4.40}_{\pm 0.60}$ |

| METHOD | RARE-OBSERVATION | | | | RARE-OBSERVATION (DIFFGROUPS) | | | |
|---|---|---|---|---|---|---|---|---|
| | AUP↑ | AUR↑ | $I_m/10^4$↑ | $S_m/10^2$↓ | AUP↑ | AUR↑ | $I_m/10^4$↑ | $S_m/10^2$↓ |
| FO | $\mathbf{1.00}_{\pm 0.00}$ | $0.13_{\pm 0.00}$ | $0.46_{\pm 0.00}$ | $47.39_{\pm 0.16}$ | $\mathbf{1.00}_{\pm 0.00}$ | $0.14_{\pm 0.00}$ | $0.50_{\pm 0.01}$ | $52.13_{\pm 0.96}$ |
| AFO | $\mathbf{1.00}_{\pm 0.00}$ | $0.16_{\pm 0.00}$ | $0.55_{\pm 0.01}$ | $56.81_{\pm 0.39}$ | $\mathbf{1.00}_{\pm 0.00}$ | $0.16_{\pm 0.01}$ | $0.54_{\pm 0.02}$ | $56.92_{\pm 1.24}$ |
| IG | $\mathbf{1.00}_{\pm 0.00}$ | $0.13_{\pm 0.00}$ | $0.46_{\pm 0.00}$ | $47.82_{\pm 0.15}$ | $\mathbf{1.00}_{\pm 0.00}$ | $0.13_{\pm 0.00}$ | $0.47_{\pm 0.00}$ | $49.90_{\pm 0.88}$ |
| SVS | $\mathbf{1.00}_{\pm 0.00}$ | $0.13_{\pm 0.00}$ | $0.46_{\pm 0.00}$ | $47.39_{\pm 0.16}$ | $\mathbf{1.00}_{\pm 0.00}$ | $0.13_{\pm 0.00}$ | $0.47_{\pm 0.01}$ | $49.53_{\pm 0.84}$ |
| DYNAMASK | $\underline{0.97}_{\pm 0.00}$ | $0.65_{\pm 0.00}$ | $8.32_{\pm 0.06}$ | $22.87_{\pm 0.58}$ | $\underline{0.98}_{\pm 0.00}$ | $0.52_{\pm 0.01}$ | $6.12_{\pm 0.10}$ | $\underline{30.88}_{\pm 0.70}$ |
| EXTRMASK | $\mathbf{1.00}_{\pm 0.00}$ | $\underline{0.76}_{\pm 0.00}$ | $\underline{13.25}_{\pm 0.07}$ | $\underline{9.55}_{\pm 0.39}$ | $\mathbf{1.00}_{\pm 0.00}$ | $\underline{0.70}_{\pm 0.04}$ | $\underline{10.40}_{\pm 0.54}$ | $32.81_{\pm 0.88}$ |
| CONTRALSP | $\mathbf{1.00}_{\pm 0.00}$ | $\mathbf{1.00}_{\pm 0.00}$ | $\mathbf{20.68}_{\pm 0.03}$ | $\mathbf{0.32}_{\pm 0.16}$ | $\mathbf{1.00}_{\pm 0.00}$ | $\mathbf{0.99}_{\pm 0.00}$ | $\mathbf{20.51}_{\pm 0.07}$ | $\mathbf{0.57}_{\pm 0.20}$ |

### 4.3 LEARNING OBJECTIVE

In our method, we utilize the preservation game (Fong & Vedaldi, 2017), where the aim is to maximize data masking while minimizing the deviation of predictions from the original ones. Thus, the overall learning objective is to train the whole framework by minimizing the total loss

$$\underset{\boldsymbol{\mu}, \theta_1, \theta_2}{\arg\min} \mathcal{L}\left(f(\boldsymbol{x}), f \circ \Phi(\boldsymbol{x}, \boldsymbol{m})\right) + \frac{\alpha}{N} \sum_{i=1}^{N} \mathcal{R}(\boldsymbol{x}_i, \boldsymbol{m}_i) + \frac{\beta}{N} \sum_{i=1}^{N} \mathcal{L}_{cntr}(\boldsymbol{x}_i), \tag{7}$$

where $\{\boldsymbol{\mu}, \theta_1, \theta_2\}$ are learnable parameters of the whole framework and $\{\alpha, \beta\}$ are hyperparameters adjusting the weight of losses to learn the sparse masks. Note that during the inference phase, we remove the random noises $\boldsymbol{\epsilon}_i$ from the sparse gates and set $\boldsymbol{m}_i = \min(1, \max(0, \boldsymbol{\mu}_i \odot \sigma(\tau(\boldsymbol{x}_i)\boldsymbol{\mu}_i))$ for deterministic masks. We summarize the pseudo-code of the proposed ContraLSP in Appendix C.

## 5 EXPERIMENTS

In this section, we evaluate the explainability of the proposed method on synthetic datasets (where truth feature importance is accessible) for both regression (white-box) and classification (black-box), as well as on more intricate real-world clinical tasks. For black-box and real-world experiments, we use 1-layer GRU with 200 hidden units as the target model $f_\theta$ to explain. All performance results for our method, benchmarks, and ablations are reported using mean $\pm$ std of 5 repetitions. For each metric in the results, we use ↑ to indicate a preference for higher values and ↓ to indicate a preference for lower values, and we mark **bold** as the best and underline as the second best. More details of each dataset and experiment are provided in Appendix D.

### 5.1 WHITE-BOX REGRESSION SIMULATION

**Datasets and Benchmarks.** Following Crabbé & Van Der Schaar (2021), we apply sparse white-box regressors whose predictions depend only on the known sub-features $\mathcal{S} = \mathcal{S}^T \times \mathcal{S}^D \subset [:, 1 : T] \times [:, 1 : D]$ as salient indices. Besides, we extend our investigation by incorporating heterogeneous samples to explore the influence of inter-samples on masking. Specifically, we consider the subset of samples from two unequal nonlinear groups $\{\mathcal{S}_1, \mathcal{S}_2\} \subset \mathcal{S}$, denoted as *DiffGroups*. Here, $\mathcal{S}_1$ and $\mathcal{S}_2$ collectively constitute the entire set $\mathcal{S}$, with each subset having a size of $|\mathcal{S}_1| = |\mathcal{S}_2| = |\mathcal{S}|/2 = 50$. The salient features are represented mathematically as

$$[f(\boldsymbol{x})]_t = \begin{cases} \sum_{[:,t,d] \in \mathcal{S}} (\boldsymbol{x}[t,d])^2 & \text{if in } \mathcal{S} \\ 0 & \text{else,} \end{cases} \quad \text{and} \quad [f(\boldsymbol{x})]_t = \begin{cases} \sum_{[i,t,d] \in \mathcal{S}_1} (\boldsymbol{x}_i[t,d])^2 & \text{if in } \mathcal{S}_1 \\ \left(\sum_{[j,t,d] \in \mathcal{S}_2} \boldsymbol{x}_j[t,d]\right)^2 & \text{elif in } \mathcal{S}_2 \\ 0 & \text{else.} \end{cases}$$

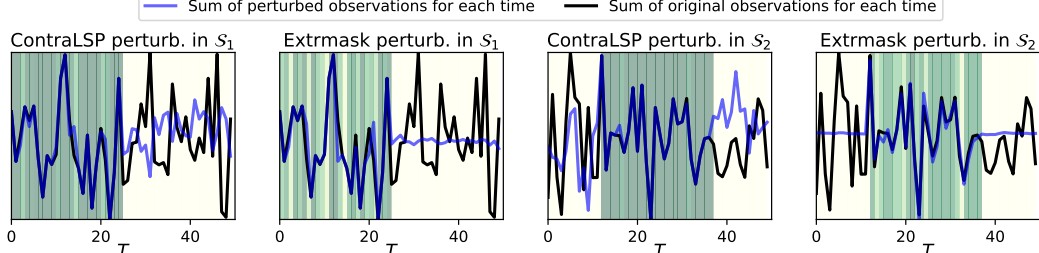

Figure 5: Differences between ContraLSP and Extrmask perturbations on the Rare-Observation (Diffgroups) experiment. We randomly select a sample in each of the two groups and sum all observations. The background color represents the mask value, with darker colors indicating higher values. ContraLSP provides counterfactual information, yet Extrmask's perturbation is close to 0.

In our experiments, we separately examine two scenarios with and without *DiffGroups*: where setting $|\mathcal{S}^T| \ll N \times T$ is called *Rare-Time* and setting $|\mathcal{S}^D| \ll N \times D$ is called *Rare-Observation*. These scenarios are recognized in saliency methods due to their inherent complexity (Ismail et al., 2019). In fact, some methods are not applicable to evaluate white-box regression models, e.g., DeepLIFT (Shrikumar et al., 2017) and FIT (Tonekaboni et al., 2020). To ensure a fair comparison, we compare ContraLSP with several baseline methods, including Feature Occlusion (FO) (Suresh et al., 2017), Augmented Feature Occlusion (AFO) (Tonekaboni et al., 2020), Integrated Gradient (IG) (Sundararajan et al., 2017), Shapley Value Sampling (SVS) (Castro et al., 2009), Dynamask (Crabbé & Van Der Schaar, 2021), and Extrmask (Enguehard, 2023). The implementation details of all algorithms are available in Appendix D.5.

**Metrics.** Since we know the exact cause, we utilize it as the ground truth important for evaluating explanations. Observations causing prediction label changes receive an explanation of 1, otherwise it is 0. To this end, we evaluate feature importance with area under precision (AUP) and area under recall (AUR). To gauge the information of the masks and the sharpness of region explanations, we also use two metrics introduced by Crabbé & Van Der Schaar (2021): the information $I_{\boldsymbol{m}}(\boldsymbol{a}) = -\sum_{[i,t,d]\in\boldsymbol{a}} \ln(1 - \boldsymbol{m}_i[t,d])$ and mask entropy $S_{\boldsymbol{m}}(\boldsymbol{a}) = -\sum_{[i,t,d]\in\boldsymbol{a}} \boldsymbol{m}_i[t,d]\ln(\boldsymbol{m}_i[t,d]) + (1 - \boldsymbol{m}_i[t,d])\ln(1 - \boldsymbol{m}_i[t,d])$, where $\boldsymbol{a}$ represents true salient features.

**Results.** Table 1 summarizes the performance results of the above regressors with rare salient features. AUP does not work as a performance discriminator in sparse scenarios. We find that for all metrics except AUP, our method significantly outperforms all other benchmarks. Moreover, ContraLSP identifies a notably larger proportion of genuinely important features in all experiments, even close to precise attribution, as indicated by the higher AUR. Note that when different groups are present within the samples, the performance of mask-based methods at the baseline significantly deteriorates, while ContraLSP remains relatively unaffected. We present a comparison between the perturbations generated by ContraLSP and Extrmask, as shown in Figure 5. This suggests that employing counterfactuals for learning contrastive inter-samples leads to less information in non-salient areas and highlights the mask more compared to other methods. We display the saliency maps for rare experiments, which are shown in the Appendix G. Our method accurately captures the important features with some smoothing in this setting, indicating that the sparse gates are working. We also explore in Appendix F whether different perturbations keep the original data distribution.

## 5.2 BLACK-BOX CLASSIFICATION SIMULATION

**Datasets and Benchmarks.** We reproduce the *Switch-Feature* and *State* experiments from Tonekaboni et al. (2020). The *Switch-Feature* data introduces complexity by altering features using a Gaussian Process (GP) mixture model. For the *State* dataset, we introduce intricate temporal dynamics using a non-stationary Hidden Markov Model (HMM) to generate multivariate altering observations with time-dependent state transitions. These alterations influence the predictive distribution, highlighting the importance of identifying key features during state transitions. Therefore, an accurate generator for capturing temporal dynamics is essential in this context. For a further description of the datasets, see Appendix D.2. For the benchmarks, in addition to the previous ones, we also use FIT, DeepLIFT, GradSHAP (Lundberg & Lee, 2017), LIME (Ribeiro et al., 2016), and RETAIN (Choi et al., 2016).

Table 2: Performance on Switch Feature and State data.

| Method | SWITCH-FEATURE | | | | STATE | | | |
|---|---|---|---|---|---|---|---|---|
| | AUP ↑ | AUR ↑ | $I_m/10^4$ ↑ | $S_m/10^3$ ↓ | AUP ↑ | AUR ↑ | $I_m/10^4$ ↑ | $S_m/10^3$ ↓ |
| FO | $0.89_{\pm 0.03}$ | $0.37_{\pm 0.02}$ | $1.86_{\pm 0.14}$ | $15.60_{\pm 0.28}$ | $0.90_{\pm 0.05}$ | $0.30_{\pm 0.01}$ | $2.73_{\pm 0.15}$ | $28.07_{\pm 0.54}$ |
| AFO | $0.82_{\pm 0.06}$ | $0.41_{\pm 0.02}$ | $2.00_{\pm 0.14}$ | $17.32_{\pm 0.29}$ | $0.84_{\pm 0.08}$ | $0.36_{\pm 0.03}$ | $3.16_{\pm 0.27}$ | $34.03_{\pm 1.10}$ |
| IG | $0.91_{\pm 0.02}$ | $0.44_{\pm 0.03}$ | $2.21_{\pm 0.17}$ | $16.87_{\pm 0.52}$ | $\underline{0.93}_{\pm 0.02}$ | $0.34_{\pm 0.03}$ | $3.17_{\pm 0.28}$ | $30.19_{\pm 1.22}$ |
| GRADSHAP | $0.88_{\pm 0.02}$ | $0.38_{\pm 0.02}$ | $1.92_{\pm 0.13}$ | $15.85_{\pm 0.40}$ | $0.88_{\pm 0.06}$ | $0.30_{\pm 0.02}$ | $2.76_{\pm 0.20}$ | $28.18_{\pm 0.96}$ |
| DEEPLIFT | $0.91_{\pm 0.02}$ | $0.44_{\pm 0.02}$ | $2.23_{\pm 0.16}$ | $16.86_{\pm 0.52}$ | $\underline{0.93}_{\pm 0.02}$ | $0.35_{\pm 0.03}$ | $3.20_{\pm 0.27}$ | $30.21_{\pm 1.19}$ |
| LIME | $0.94_{\pm 0.02}$ | $0.40_{\pm 0.02}$ | $2.01_{\pm 0.13}$ | $16.09_{\pm 0.58}$ | $\mathbf{0.95}_{\pm 0.02}$ | $0.32_{\pm 0.03}$ | $2.94_{\pm 0.26}$ | $28.55_{\pm 1.53}$ |
| FIT | $0.48_{\pm 0.03}$ | $0.43_{\pm 0.02}$ | $1.99_{\pm 0.11}$ | $17.16_{\pm 0.50}$ | $0.45_{\pm 0.02}$ | $0.59_{\pm 0.02}$ | $7.92_{\pm 0.40}$ | $33.59_{\pm 0.17}$ |
| RETAIN | $0.93_{\pm 0.01}$ | $0.33_{\pm 0.04}$ | $1.54_{\pm 0.20}$ | $15.08_{\pm 1.13}$ | $0.52_{\pm 0.16}$ | $0.21_{\pm 0.01}$ | $1.56_{\pm 0.24}$ | $25.01_{\pm 0.57}$ |
| DYNAMASK | $0.35_{\pm 0.00}$ | $\underline{0.77}_{\pm 0.02}$ | $5.22_{\pm 0.26}$ | $12.85_{\pm 0.53}$ | $0.36_{\pm 0.01}$ | $\underline{0.79}_{\pm 0.01}$ | $10.59_{\pm 0.20}$ | $25.11_{\pm 0.40}$ |
| EXTRMASK | $\underline{0.97}_{\pm 0.01}$ | $0.65_{\pm 0.05}$ | $\underline{8.45}_{\pm 0.51}$ | $\underline{6.90}_{\pm 1.44}$ | $0.87_{\pm 0.01}$ | $0.77_{\pm 0.01}$ | $\underline{29.71}_{\pm 1.39}$ | $\underline{7.54}_{\pm 0.46}$ |
| CONTRALSP | $\mathbf{0.98}_{\pm 0.00}$ | $\mathbf{0.80}_{\pm 0.03}$ | $\mathbf{24.23}_{\pm 1.27}$ | $\mathbf{0.91}_{\pm 0.26}$ | $0.90_{\pm 0.03}$ | $\mathbf{0.81}_{\pm 0.01}$ | $\mathbf{50.09}_{\pm 0.78}$ | $\mathbf{0.50}_{\pm 0.05}$ |

Table 3: Effects of contrastive perturbations (using the triplet loss) and smoothing constraint (using the trend function) on the Switch-Feature and State datasets.

| Method | SWITCH-FEATURE | | | | STATE | | | |
|---|---|---|---|---|---|---|---|---|
| | AUP ↑ | AUR ↑ | $I_m/10^4$ ↑ | $S_m/10^3$ ↓ | AUP ↑ | AUR ↑ | $I_m/10^4$ ↑ | $S_m/10^3$ ↓ |
| CONTRALSP W/O BOTH | $0.92_{\pm 0.01}$ | $\underline{0.79}_{\pm 0.02}$ | $22.08_{\pm 1.43}$ | $\underline{0.78}_{\pm 0.16}$ | $0.76_{\pm 0.02}$ | $0.74_{\pm 0.01}$ | $42.26_{\pm 0.45}$ | $\mathbf{0.14}_{\pm 0.02}$ |
| CONTRALSP W/O TRIPLET LOSS | $\underline{0.97}_{\pm 0.01}$ | $\underline{0.79}_{\pm 0.02}$ | $22.99_{\pm 0.84}$ | $1.00_{\pm 0.21}$ | $\underline{0.88}_{\pm 0.03}$ | $\underline{0.80}_{\pm 0.01}$ | $\underline{49.04}_{\pm 0.75}$ | $0.76_{\pm 0.07}$ |
| CONTRALSP W/O TREND FUNCTION | $0.92_{\pm 0.01}$ | $\mathbf{0.80}_{\pm 0.01}$ | $\underline{24.16}_{\pm 0.69}$ | $\mathbf{0.65}_{\pm 0.10}$ | $0.77_{\pm 0.02}$ | $\underline{0.80}_{\pm 0.01}$ | $42.22_{\pm 0.50}$ | $\underline{0.15}_{\pm 0.02}$ |
| CONTRALSP | $\mathbf{0.98}_{\pm 0.00}$ | $\mathbf{0.80}_{\pm 0.03}$ | $\mathbf{24.23}_{\pm 1.27}$ | $0.91_{\pm 0.26}$ | $\mathbf{0.90}_{\pm 0.03}$ | $\mathbf{0.81}_{\pm 0.01}$ | $\mathbf{50.09}_{\pm 0.78}$ | $0.50_{\pm 0.05}$ |

**Metrics.** We maintain consistency with the ones previously employed.

**Results.** The performance results on simulated data are presented in Table 2. Across Switch-Feature and State settings, ContraLSP is the best explainer on 7/8 (4 metrics in two datasets) over the strongest baselines. Specifically, when AUP is at the same level, our method achieves high AUR results from its emphasis on producing smooth masks over time, favoring complete subsequence patterns over sparse portions, aligning with human interpretation needs. The reason why Dynamask has a high AUR is that the failure pro-

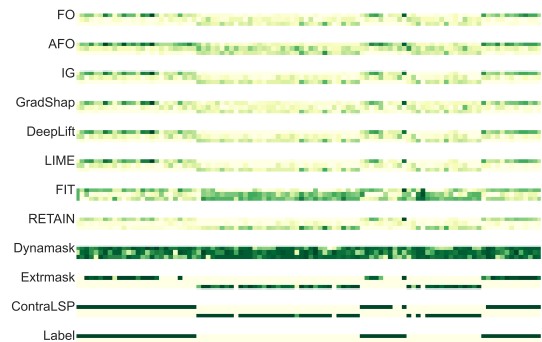

Figure 6: Saliency maps produced by various methods for Switch-Feature data.

duces a smaller region of masks, as shown in Figure 6. ContraLSP also has an average 94.75% improvement in the information content $I_m$ and an average 90.24% reduction in the entropy $S_m$ over the strongest baselines. This indicates that the contrastive perturbation is superior to perturbation by other means when explaining forecasts based on multivariate time series data.

**Ablation study.** We further explore these two datasets with the ablation study of two crucial components of the model: (i) let $\mathcal{L}_{cntr}$ to cancel contrastive learning with the triplet loss and (ii) without the trend function $\tau_{\theta_2}(\cdot)$ so that $\boldsymbol{\mu}' = \boldsymbol{\mu}$. As shown in Table 3, the ContraLSP with both components performs best. Whereas without the use of triplet loss, the performance degrades as the method fails to learn the mask with counterfactuals. Such perturbations without contrastive optimization are not sufficiently uninformative, leading to a lack of distinction among samples. Moreover, equipped with the trend function, ContraLSP improves the AUP by 0.06 and 0.13 on the two datasets, respectively. It indicates that temporal trends introduce context as a smoothing factor, which improves the explanatory ability of our method. To determine the values of $\alpha$ and $\beta$ in Eq. (7), we also show different values for parameter combination, which are given in more detail in the Appendix E.2.

## 5.3 MIMIC-III MORTALITY DATA

**Dataset and Benchmarks.** We use the MIMIC-III dataset (Johnson et al., 2016), which is a comprehensive clinical time series dataset encompassing various vital and laboratory measurements. It is extensively utilized in healthcare and medical artificial intelligence-related research. For more details, please refer to Appendix D.3. We use the same benchmarks as before the classification.

Table 4: Performance report on MIMIC-III mortality by masking 20% data.

| METHOD | AVERAGE SUBSTITUTION | | | | ZERO SUBSTITUTION | | | |
| --- | --- | --- | --- | --- | --- | --- | --- | --- |
| | ACC ↓ | CE ↑ | SUFF$*10^2$ ↓ | COMP$*10^2$ ↑ | ACC ↓ | CE ↑ | SUFF$*10^2$ ↓ | COMP$*10^2$ ↑ |
| FO | $0.988_{\pm0.001}$ | $0.094_{\pm0.005}$ | $0.455_{\pm0.076}$ | $-0.229_{\pm0.059}$ | $0.971_{\pm0.003}$ | $0.121_{\pm0.008}$ | $-0.539_{\pm0.169}$ | $-0.523_{\pm0.274}$ |
| AFO | $0.989_{\pm0.002}$ | $0.097_{\pm0.005}$ | $0.185_{\pm0.122}$ | $0.008_{\pm0.077}$ | $0.972_{\pm0.004}$ | $0.120_{\pm0.008}$ | $-0.546_{\pm0.322}$ | $-0.169_{\pm0.240}$ |
| IG | $0.988_{\pm0.002}$ | $0.096_{\pm0.005}$ | $0.273_{\pm0.098}$ | $-0.080_{\pm0.150}$ | $0.971_{\pm0.004}$ | $0.122_{\pm0.006}$ | $-0.474_{\pm0.228}$ | $-0.385_{\pm0.268}$ |
| GRADSHAP | $0.987_{\pm0.003}$ | $0.095_{\pm0.005}$ | $0.400_{\pm0.103}$ | $-0.219_{\pm0.058}$ | $0.968_{\pm0.005}$ | $0.128_{\pm0.015}$ | $0.066_{\pm0.460}$ | $-0.628_{\pm0.377}$ |
| DEEPLIFT | $0.987_{\pm0.002}$ | $0.095_{\pm0.004}$ | $0.303_{\pm0.104}$ | $-0.115_{\pm0.140}$ | $0.972_{\pm0.004}$ | $0.119_{\pm0.004}$ | $-0.427_{\pm0.193}$ | $-0.482_{\pm0.246}$ |
| LIME | $0.997_{\pm0.001}$ | $0.094_{\pm0.005}$ | $0.116_{\pm0.122}$ | $-0.028_{\pm0.050}$ | $0.988_{\pm0.003}$ | $0.099_{\pm0.004}$ | $1.688_{\pm0.472}$ | $0.254_{\pm0.241}$ |
| FIT | $0.996_{\pm0.01}$ | $0.098_{\pm0.004}$ | $-0.139_{\pm0.139}$ | $0.375_{\pm0.067}$ | $0.987_{\pm0.004}$ | $0.108_{\pm0.07}$ | $-0.745_{\pm0.450}$ | $1.053_{\pm0.224}$ |
| RETAIN | $0.988_{\pm0.001}$ | $0.092_{\pm0.005}$ | $0.788_{\pm0.046}$ | $-0.425_{\pm0.096}$ | $0.971_{\pm0.004}$ | $0.119_{\pm0.008}$ | $0.072_{\pm0.394}$ | $-0.984_{\pm0.266}$ |
| DYNAMASK | $0.990_{\pm0.001}$ | $0.099_{\pm0.005}$ | $-0.083_{\pm0.089}$ | $0.354_{\pm0.064}$ | $0.976_{\pm0.004}$ | $0.114_{\pm0.007}$ | $-0.422_{\pm0.501}$ | $0.609_{\pm0.170}$ |
| EXTRMASK | $\underline{0.982}_{\pm0.003}$ | $\underline{0.118}_{\pm0.007}$ | $\underline{-1.157}_{\pm0.362}$ | $\underline{1.538}_{\pm0.395}$ | $\underline{0.943}_{\pm0.007}$ | $\underline{0.318}_{\pm0.051}$ | $\mathbf{-6.942}_{\pm0.531}$ | $\underline{10.847}_{\pm2.055}$ |
| CONTRALSP | $\mathbf{0.980}_{\pm0.002}$ | $\mathbf{0.127}_{\pm0.007}$ | $\mathbf{-1.792}_{\pm0.095}$ | $\mathbf{2.386}_{\pm0.175}$ | $\mathbf{0.928}_{\pm0.020}$ | $\mathbf{0.357}_{\pm0.044}$ | $\underline{-6.636}_{\pm0.315}$ | $\mathbf{17.442}_{\pm2.544}$ |

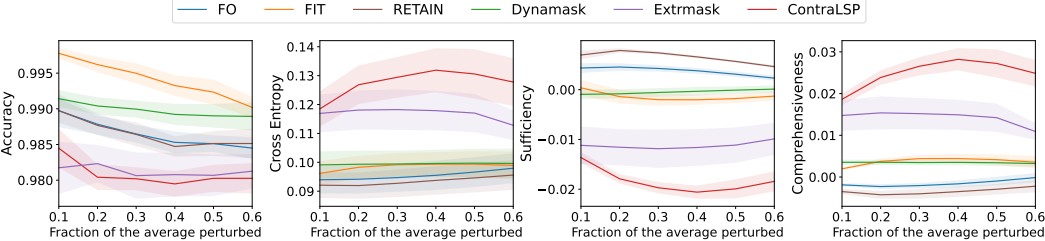

Figure 7: Quantitative results on the MIMIC-III mortality experiment, focusing on Accuracy ↓, Cross Entropy ↑, Sufficiency ↓, and Comprehensiveness ↑. We mask a varying percentage of the data (ranging from 10% to 60%) for each patient and replace the masked data with the overall average over time for each feature: $\overline{\boldsymbol{x}}_i[t,d] = \sum_{t=1}^{T} \boldsymbol{x}_i[t,d]$. Since some curves are similar, we show representative baselines for clarity.

**Metrics.** Due to the absence of real attribution features in MIMIC-III, we mask certain portions of the features to assess their importance. We report that performance is evaluated using top mask substitution, as is done in Enguehard (2023). It replaces masked features either with an average over time of this feature ($\overline{\boldsymbol{x}}_i[t,d] = \frac{1}{T}\sum_{t=1}^{T} \boldsymbol{x}_i[t,d]$) or with zeros ($\overline{\boldsymbol{x}}_i[t,d] = 0$). The metrics we select are Accuracy (Acc, lower is better), Cross-Entropy (CE, higher is better), Sufficiency (Suff, lower is better), and Comprehensiveness (Comp, higher is better), where the details are in Appendix D.3.

**Results.** The performance results on MIMIC-III mortality by masking 20% data are presented in Table 4. We can see that our method outperforms the leading baseline Extrmask on 7/8 metrics (across 4 metrics in two substitutions). Compared to other methods on feature-removal (FO, AFO, FIT) and gradient (IG, DeepLift, GradShap), the gains are greater. The reason could be that the local mask produced by ContraLSP is sparser than others and is replaced by more uninformative perturbations. We show the details of hyperparameter determination for the MIMIC-III dataset, which is deferred to Appendix E.2. Considering that replacement masks different proportions of the data, we also show the average substitution using the above metrics in Figure 7, where 10% to 60% of the data is masked for each patient. Our results show that our method outperforms others in most cases. This indicates that perturbations using contrastive learning are superior to those using other perturbations in interpreting forecasts for multivariate time series data.

## 6 CONCLUSION

We introduce ContraLSP, a perturbation-base model designed for the interpretation of time series models. By incorporating counterfactual samples and sample-specific sparse gates, ContraLSP not only offers contractive perturbations but also maintains sparse salient areas. The smooth constraint applied through temporal trends further enhances the model's ability to align with latent patterns in time series data. The performance of ContraLSP across various datasets and its ability to reveal essential patterns make it a valuable tool for enhancing the transparency and interpretability of time series models in diverse fields. However, generating perturbations by the contrasting objective may not bring counterfactuals strong enough, since it is label-free generation. Besides, an inherent limitation of our method is the selection of sparse parameters, especially when dealing with different datasets. Addressing this challenge may involve the implementation of more parameter-efficient tuning strategies, so it would be interesting to explore one of these adaptations to salient areas.

ACKNOWLEDGMENTS

This work was supported by Alibaba Group through Alibaba Research Intern Program.

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

## A    REGULARIZATION TERM

Let $\mathrm{erf}$ be the Gaussian error function defined as $\mathrm{erf}(x) = \frac{2}{\sqrt{\pi}} \int_0^x e^{-t^2} dt$, and let the mask $\boldsymbol{m}_i$ be obtained with the sigmoid gate output $\boldsymbol{\mu}_i'$ and an injected noise $\boldsymbol{\epsilon}_i$ from $\mathcal{N}(0, \delta^2)$. Thus, the regularization term for each sample $\mathcal{R}^{(i)}$ can be expressed by

$$
\begin{aligned}
\mathcal{R}^{(i)}(\boldsymbol{x}_i, \boldsymbol{m}_i) &= \mathbb{E}\left[\alpha \left\|\boldsymbol{m}_i\right\|_0\right] \\
&= \alpha \sum_{t=1}^T \sum_{d=1}^D \mathbb{P}\left(\boldsymbol{\mu}_i'[t,d] + \boldsymbol{\epsilon}_i[t,d] > 0\right) \\
&= \alpha \sum_{t=1}^T \sum_{d=1}^D \left[1 - \mathbb{P}\left(\boldsymbol{\mu}_i'[t,d] + \boldsymbol{\epsilon}_i[t,d] \leq 0\right)\right] \\
&= \alpha \sum_{t=1}^T \sum_{d=1}^D \left[1 - \Psi\left(\frac{-\boldsymbol{\mu}_i'[t,d]}{\delta}\right)\right] \\
&= \alpha \sum_{t=1}^T \sum_{d=1}^D \Psi\left(\frac{\boldsymbol{\mu}_i'[t,d]}{\delta}\right) \\
&= \alpha \sum_{t=1}^T \sum_{d=1}^D \left(\frac{1}{2} - \frac{1}{2}\mathrm{erf}(-\frac{\boldsymbol{\mu}_i'[t,d]}{\sqrt{2}\delta})\right) \\
&= \alpha \sum_{t=1}^T \sum_{d=1}^D \left(\frac{1}{2} + \frac{1}{2}\mathrm{erf}(\frac{\boldsymbol{\mu}_i'[t,d]}{\sqrt{2}\delta})\right),
\end{aligned}
\tag{8}
$$

where $\Psi(\cdot)$ is the cumulative distribution function, and $\boldsymbol{\mu}_i'[t,d]$ is computed by Eq. (4).

## B    TRIPLE SAMPLES SELECTED

In this section, we describe how to generate positive and negative samples for contrastive learning. For each sample $\boldsymbol{x}_i$, our goal is to generate the counterfactuals $\boldsymbol{x}_i^r$ via the perturbation function $\varphi(\cdot)$, optimized to be counterfactual for an uninformative perturbed sample. The pseudo-code of the triplet sample selection is shown in Algorithm 1 and elaborated as follows. (i) We start by clustering samples in each batch into the positives $\Omega^+$ and the negatives $\Omega^-$ with 2-kmeans, (ii) we select the current sample from each cluster as an anchor, along with $K^+$ nearest samples from the same cluster as the positive samples, (iii) and we select $K^-$ random samples from the other cluster yielding negative samples. Note that we use $S_p$ and $S_n$ as auxiliary variables representing two sets to select positive and negative samples, respectively.

---

**Algorithm 1** Selection of a triplet sample

---

**Input:** The set of perturbation time series $\Omega = \{\boldsymbol{x}_i^r\}_{i=1}^N$ and the current perturbation $\boldsymbol{x}_i^r$.
**Output:** Triple sample $\mathcal{T}_i = (\boldsymbol{x}_i^r, \{\boldsymbol{x}_{i,k}^{r^+}\}_{k=1}^{K^+}, \{\boldsymbol{x}_{i,k}^{r^-}\}_{k=1}^{K^-})$
Initialize a positive set $S_p = \{\}$ and a negative set $S_n = \{\}$
Clustering positive and negative samples $\{\Omega^+, \Omega^-\} \leftarrow$ 2-kmeans($\Omega$)
**for** $\Omega^*$ in $\{\Omega^+, \Omega^-\}$ **do**
   Select Anchor $\boldsymbol{x}_i^r \in \Omega^*$
   $\Omega^* \leftarrow \Omega^* \setminus \{\boldsymbol{x}_i^r\}$
   **for** $k \leftarrow 1$ to $K^+$ **do**
      $\boldsymbol{x}_{i,k}^{r^+} = \Omega^*.\mathrm{Top}(\boldsymbol{x}_i^r)$
      $\Omega^* \leftarrow \Omega^* \setminus \{\boldsymbol{x}_{i,k}^{r^+}\}, S_p \leftarrow S_p \cup \boldsymbol{x}_{i,k}^{r^+}$
   **end for**
   **for** $k \leftarrow 1$ to $K^-$ **do**
      $\boldsymbol{x}_{i,k}^{r^-} = \mathrm{random}(\Omega \setminus \Omega^*)$
      $\Omega^* \leftarrow \Omega^* \setminus \{\boldsymbol{x}_{i,k}^{r^-}\}, S_n \leftarrow S_n \cup \boldsymbol{x}_{i,k}^{r^-}$
   **end for**
**end for**
**Output:** Triple sample $(\boldsymbol{x}_i^r, \{\boldsymbol{x}_{i,k}^{r^+}\}_{k=1}^{K^+}, \{\boldsymbol{x}_{i,k}^{r^-}\}_{k=1}^{K^-})$

---

## C  PSEUDO CODE

---

**Algorithm 2** The pseudo-code of our ContraLSP

---

**Input:** Multi-variate time series $\{\boldsymbol{x}_i\}_{i=1}^N$, black-box model $f$, sparsity hyper-parameters $\{\alpha, \beta\}$, Gaussian noise $\delta$, total training epochs $E$, learning rate $\gamma$
**Output:** Masks $\boldsymbol{m}$ to explain
**Training:**
Initialize the indicator vectors $\boldsymbol{\mu} = \{\boldsymbol{\mu}_i\}_{i=1}^N$ of sparse perturbation
Initialize a perturbation function $\varphi_{\theta_1}(\cdot)$ and a trend function $\tau_{\theta_2}(\cdot)$
**for** $e \leftarrow 1$ to $E$ **do**
  **for** $i \leftarrow 1$ to $N$ **do**
    Get time treads $\{\tau_{\theta_2}(\boldsymbol{x}_i[:, d])\}_{d=1}^D$ in each observations $\boldsymbol{x}_i[:, d]$
    Compute $\boldsymbol{\mu}'_i \leftarrow \boldsymbol{\mu}_i \odot \sigma(\tau(\boldsymbol{x}_i)\boldsymbol{\mu}_i)$
    Sample $\boldsymbol{\epsilon}_i$ from the Gaussian distribution $\mathcal{N}(0, \delta)$
    Compute instance-wise masks $\boldsymbol{m}_i \leftarrow \min(1, \max(0, \boldsymbol{\mu}'_i + \boldsymbol{\epsilon}_i))$
    Get counterfactual features $\boldsymbol{x}_i^r \leftarrow \varphi_{\theta_1}(\boldsymbol{x}_i)$
    Compute the triplet loss $\mathcal{L}_{cntr}$ via Alg. 1 and Eq. (2)
    Compute the regularization term $\mathcal{R}(\boldsymbol{x}_i, \boldsymbol{m}_i)$ via Eq. (5)
  **end for**
  Get perturbations $\Phi(\boldsymbol{x}, \boldsymbol{m}) \leftarrow \boldsymbol{m} \odot \boldsymbol{x} + (\boldsymbol{1} - \boldsymbol{m}) \odot \boldsymbol{x}^r$
  Construct the total loss function:
    $\widetilde{\mathcal{L}} = \mathcal{L}(f(\boldsymbol{x}), f \circ \Phi(\boldsymbol{x}, \boldsymbol{m})) + \frac{\alpha}{N}\sum_{i=1}^N \mathcal{R}(\boldsymbol{x}_i, \boldsymbol{m}_i) + \frac{\beta}{N}\sum_{i=1}^N \mathcal{L}_{cntr}(\boldsymbol{x}_i)$
  Update $\boldsymbol{\mu} \leftarrow \boldsymbol{\mu} - \gamma\nabla_{\boldsymbol{\mu}}\widetilde{\mathcal{L}}, \; \theta_1 \leftarrow \theta_1 - \gamma\nabla_{\theta_1}\widetilde{\mathcal{L}}, \; \theta_2 \leftarrow \theta_2 - \gamma\nabla_{\theta_2}\widetilde{\mathcal{L}}$
**end for**
Store $\boldsymbol{\mu}, \varphi_{\theta_1}(\cdot), \tau_{\theta_2}(\cdot)$
**Inference:** Compute final masks $\boldsymbol{m} \leftarrow \min(1, \max(0, \boldsymbol{\mu} \odot \sigma(\tau(\boldsymbol{x})\boldsymbol{\mu})))$
**Return:** Masks $\boldsymbol{m}$

---

## D  EXPERIMENTAL SETTINGS AND DETAILS

### D.1  WHITE-BOX REGRESSION DATA

As this experiment relies on a white-box approach, our sole responsibility is to create the input sequences. As detailed by Crabbé & Van Der Schaar (2021), each feature sequence is generated using an ARMA process:

$$\boldsymbol{x}_i[t, d] = 0.25\boldsymbol{x}_i[t-1, d] + 0.1\boldsymbol{x}_i[t-2, d] + 0.05\boldsymbol{x}_i[t-3, d] + \epsilon'_i, \tag{9}$$

where $\epsilon'_i \sim \mathcal{N}(0, 1)$. We generate 100 sequence samples for each observation $d$ within the range of $d \in [1 : 50]$ and time $t$ within the range of $t \in [1 : 50]$, and set the sample size $|\mathcal{S}_1| = |\mathcal{S}_1| = |\mathcal{S}|/2 = 50$ in different group experiments.

In the experiment involving Rare-Time, we identify 5 time steps as salient in each sample, where consecutive time steps are randomly selected and differently for different groups. The salient observation instances are defined as $\mathcal{S}^D = [:, 13 : 38]$ without different groups and as $\mathcal{S}_1^D = [:, 1 : 25], \mathcal{S}_2^D = [:, 13 : 38]$ with different groups.

In the experiment involving Rare-Observation, we identify 5 salient observations in each sample without replacement from $[1 : 50]$, whereas in different groups $\mathcal{S}_1^D$ and $\mathcal{S}_2^D$ are 5 different observations randomly selected respectively. The salient time instances are defined as $\mathcal{S}^T = [:, 13 : 38]$ without different groups, and as $\mathcal{S}_1^T = [:, 1 : 25], \mathcal{S}_2^T = [:, 13 : 38]$ with different groups.

### D.2  BLACK-BOX CLASSIFICATION DATA

**Data generation on the Switch-Feature experiment.** We generate this dataset closely following Tonekaboni et al. (2020), where the time series states are generated via a two-state HMM with equal

initial state probabilities of $[\frac{1}{3}, \frac{1}{3}, \frac{1}{3}]$ and the following transition probabilities

$$\begin{bmatrix} 0.95 & 0.02 & 0.03 \\ 0.02 & 0.95 & 0.03 \\ 0.03 & 0.02 & 0.95 \end{bmatrix}.$$

The emission probability is a GP mixture, which is governed by an RBF kernel with 0.2 and uses means $\mu_1 = [0.8, -0.5, -0.2], \mu_2 = [0.0, -1.0, 0.0], \mu_3 = [-0.2, -0.2, 0.8]$ in each state. The output $y_i$ at every step is designed as

$$\boldsymbol{p}_i[t] = \begin{cases} \frac{1}{1+e^{-\boldsymbol{x}_i[t,1]}}, & \text{if } \boldsymbol{s}_i[t] = 0 \\ \frac{1}{1+e^{-\boldsymbol{x}_i[t,2]}}, & \text{elif } \boldsymbol{s}_i[t] = 1 \ , \quad \text{and} \quad y_i[t] \sim \text{Bernoulli}(\boldsymbol{p}_i[t]), \\ \frac{1}{1+e^{-\boldsymbol{x}_i[t,3]}}, & \text{elif } \boldsymbol{s}_i[t] = 2 \end{cases}$$

where $\boldsymbol{s}_i[t]$ is a single state at each time that controls the contribution of a single feature to the output, and we set 100 states: $t \in [1 : 100]$. We generate 1000 time series samples using this approach. Then we employ a single-layer GRU trained using the Adam optimizer with a learning rate of $10^{-4}$ for 50 epochs to predict $y_i$ based on $\boldsymbol{x}_i$.

**Data generation on the State experiment.** We generate this dataset following Tonekaboni et al. (2020) and Enguehard (2023). The random states of the time series are generated using a two-state HMM with $\pi = [0.5, 0.5]$ and the following transition probabilities

$$\begin{bmatrix} 0.1 & 0.9 \\ 0.1 & 0.9 \end{bmatrix}.$$

The emission probability is a multivariate Gaussian, where means are $\mu_1 = [0.1, 1.6, 0.5]$ and $\mu_2 = [-0.1, -0.4, -1.5]$. The label $y_i[t]$ is generated only using the last two observations, while the first one is irrelevant. Thus, the output $y_i$ at every step is defined as

$$\boldsymbol{p}_i[t] = \begin{cases} \frac{1}{1+e^{-\boldsymbol{x}_i[t,1]}} & \text{if } \boldsymbol{s}_i[t] = 0 \\ \frac{1}{1+e^{-\boldsymbol{x}_i[t,2]}} & \text{elif } \boldsymbol{s}_i[t] = 1 \end{cases}, \quad \text{and} \quad y_i[t] \sim \text{Bernoulli}(\boldsymbol{p}_i[t]),$$

where $\boldsymbol{s}_i[t]$ is either 0 or 1 at each time, and we generate 200 states: $t \in [1 : 200]$. We also generate 1000 time series samples using this approach and employ a single-layer GRU with 200 units trained by the Adam optimizer with a learning rate of $10^{-4}$ for 50 epochs to predict $y_i$ based on $\boldsymbol{x}_i$.

### D.3 MIMIC-III DATA

For this experiment, we opt for adult ICU admission data sourced from the MIMIC-III dataset (Johnson et al., 2016). The objective is to predict in-hospital mortality of each patient based on 48 hours of data ($T = 48$), and we need to explain the prediction model (the true salient features are unknown). For each patient, we used features and data processing consistent with Tonekaboni et al. (2020). We summarize all the observations in Table 5, with a total of $D = 31$. Patients with complete 48-hour blocks missing for specific features are excluded, resulting in 22,988 ICU admissions. The predicted model we train is a single-layer RNN consisting of 200 GRU cells. It undergoes training for 80 epochs using the Adam optimizer with a learning rate of 0.001.

Table 5: List of clinical observations at each time for the risk predictor model.

| DATA CLASS | NAME |
|---|---|
| STATIC OBSERVATIONS | AGE, GENDER, ETHNICITY, FIRST ADMISSION TO THE ICU |
| LAB OBSERVATIONS | LACTATE, MAGNESIUM, PHOSPHATE, PLATELET, |
| | POTASSIUM, PTT, INR, PT, SODIUM, BUN, WBC |
| VITAL OBSERVATIONS | HEARTRATE, DIASBP, SYSBP, MEANBP, RESPRATE, SPO2, GLUCOSE, TEMP |

In this task, we introduce the same metrics as Enguehard (2023), which are detailed as follows: (i) Accuracy (Acc) means the prediction accuracy while salient features selected by the model are removed, so a lower value is preferable. (ii) Cross-Entropy (CE) represents the entropy between

Table 6: Experimental settings for ContraLSP across all datasets.

| PARAMETER | RATE-TIME | RATE-OBSERVATION | SWITCH-FEATURE | STATE | MIMIC-III |
|---|---|---|---|---|---|
| LEARNING RATE $\gamma$ | 0.1 | 0.1 | 0.01 | 0.01 | 0.1 |
| OPTIMIZER | ADAM | ADAM | ADAM | ADAM | ADAM |
| MAX EPOCHS $E$ | 200 | 200 | 500 | 500 | 200 |
| $\alpha$ | 0.1 | 0.1 | 1.0 | 2.0 | 0.005 |
| $\beta$ | 0.1 | 0.1 | 2.0 | 1.0 | 0.01 |
| $\delta$ | 0.5 | 0.5 | 0.8 | 0.5 | 0.5 |
| $K^+$ | $|\Omega^+|/5$ | $|\Omega^+|/5$ | $|\Omega^+|/5$ | $|\Omega^+|/5$ | 50 |
| $K^-$ | $|\Omega^-|/5$ | $|\Omega^-|/5$ | $|\Omega^-|/5$ | $|\Omega^-|/5$ | 50 |

Table 7: The specific structure of the trend function.

| NO. | STRUCTURE |
|---|---|
| 1ST OBS. | MLP[LINEAR($T$, 32), RELU, LINEAR(32, $T$)] |
| 2ND OBS. | MLP[LINEAR($T$, 32), RELU, LINEAR(32, $T$)] |
| . . . | . . . |
| $D$TH OBS. | MLP[LINEAR($T$, 32), RELU, LINEAR(32, $T$)] |

the predictions of perturbed features with the original features. It quantifies the information loss when crucial features are omitted, with a higher value being preferable. (iii) Sufficiency (Suff) is the average change in predicted class probabilities relative to the original values, with lower values being preferable. (iv) Comprehensiveness (Comp) is the average difference of target class prediction probability when most salient features are removed. It reflects how much the removal of features hinders the prediction, so a higher value is better.

## D.4 DETAILS OF OUR METHOD

We list hyperparameters for each experiment performed in Table 6, and for the triplet loss, the marginal parameter $b$ is consistently set to 1. The size of $K^+$ and $K^-$ are chosen to depend on the number of positive and negative samples ($|\Omega^+|$ and $|\Omega^-|$). In the perturbation function $\varphi_{\theta_1}(\cdot)$, we use a single-layer bidirectional GRU, which corresponds to a generalization of the fixed perturbation. In the trend function $\tau_{\theta_2}(\cdot)$, we employ an independent MLP for each observation $d$ to find its trend, whose details are shown in Table 7. Please refer to our codebase[1] for additional details on these hyperparameters and implementations.

## D.5 DETAILS OF BENCHMARKS

We compare our method against ten popular benchmarks, including FO (Suresh et al., 2017), AFO (Tonekaboni et al., 2020), IG (Sundararajan et al., 2017), GradSHAP (Lundberg & Lee, 2017) (SVS (Castro et al., 2009) in regression), FIT (Tonekaboni et al., 2020), DeepLIFT (Shrikumar et al., 2017), LIME (Shrikumar et al., 2017), RETAIN (Choi et al., 2016), Dynamask (Crabbé & Van Der Schaar, 2021), and Extrmask (Enguehard, 2023), whereas the implementation of benchmarks is based on open source codes time_interpret[2] and DynaMask[3]. All hyperparameters follow the code provided by the authors.

## E ADDITIONAL ABLATION STUDY

### E.1 EFFECT OF DISTANCE TYPE IN CONTRASTIVE LEARNING.

For the instance-wise similarity, we can consider various losses to maximize the distance between the anchor with positive or negative samples in Eq. (2). We evaluate three typical distance metrics in Rare-Time and Rare-Observation datasets: Manhattan distance, Euclidean distance, and cosine

---

[1] https://github.com/zichuan-liu/ContraLSP
[2] https://github.com/josephenguehard/time_interpret
[3] https://github.com/JonathanCrabbe/Dynamask

Table 8: Performance of ContraLSP with different contrastive loss types on rare experiments.

| DISTANCE TYPE IN $\mathcal{L}_{cntr}$ | RARE-TIME | | | | RARE-OBSERVATION | | | |
| --- | --- | --- | --- | --- | --- | --- | --- | --- |
| | AUP ↑ | AUR ↑ | $I_m/10^4$ ↑ | $S_m/10^2$ ↓ | AUP ↑ | AUR ↑ | $I_m/10^4$ ↑ | $S_m/10^2$ ↓ |
| MANHATTAN DISTANCE | $\mathbf{1.00}_{\pm 0.00}$ | $\mathbf{0.97}_{\pm 0.01}$ | $19.51_{\pm 0.30}$ | $\mathbf{4.65}_{\pm 0.71}$ | $\mathbf{1.00}_{\pm 0.00}$ | $\mathbf{1.00}_{\pm 0.00}$ | $20.68_{\pm 0.03}$ | $\mathbf{0.32}_{\pm 0.16}$ |
| EUCLIDEAN DISTANCE | $\mathbf{1.00}_{\pm 0.00}$ | $0.97_{\pm 0.02}$ | $\mathbf{19.67}_{\pm 0.52}$ | $4.97_{\pm 0.55}$ | $\mathbf{1.00}_{\pm 0.00}$ | $1.00_{\pm 0.01}$ | $\mathbf{20.72}_{\pm 0.06}$ | $0.69_{\pm 0.17}$ |
| COSINE SIMILARITY | $\mathbf{1.00}_{\pm 0.00}$ | $0.96_{\pm 0.02}$ | $18.41_{\pm 0.64}$ | $5.87_{\pm 0.74}$ | $\mathbf{1.00}_{\pm 0.00}$ | $0.98_{\pm 0.01}$ | $19.22_{\pm 0.06}$ | $0.98_{\pm 0.23}$ |

similarity. The results presented in Table 8 indicate that the Manhattan distance is slightly better than the other evaluated losses.

### E.2 EFFECT OF REGULARIZATION FACTOR.

We conduct ablations on the black-box classification data using our method to determine which values of $\alpha$ and $\beta$ should be used in Eq. (7). For each parameter combination, we employed five distinct seeds, and the experimental results for Switch-Feature and State are presented in Table 9 and Table 10, respectively. Higher values of AUP and AUR are preferred, and the underlined values represent the best parameter pair associated with these metrics. Those Tables indicate that the $\ell$-regularized mask $\boldsymbol{m}$ is most appropriate when $\alpha$ is set to 1.0 and 2.0 for both Switch-Feature and State data, allowing for the retention of a small but highly valuable subset of features. Moreover, to force $\varphi_{\theta_1}(\cdot)$ to learn counterfactual perturbations from other distinguishable samples, $\beta$ is best set to 2.0 and 1.0, respectively. Otherwise, the perturbation may contain crucial features of the current sample, thereby impacting the classification.

We also perform ablation on the MIMIC-III dataset for parameters $\alpha$ and $\beta$ using our method. We employ Accuracy and Cross-Entropy as metrics and show the average substitution in Table 11. This Table shows that $\beta$ is best set to 0.01 to learn counterfactual perturbations. Note that the results are better when lower values of $\alpha$ are used, but over-regularizing $\boldsymbol{m}$ close to 0 may not be beneficial. Notably, lower values of $\alpha$ yield superior results, but excessively regularizing $\boldsymbol{m}$ toward 0 may prove disadvantageous (Enguehard, 2023). Therefore, we select $\alpha = 0.005$ and $\beta = 0.01$ as deterministic parameters on the MIMIC-III dataset.

Table 9: Effects of $\alpha$ and $\beta$ on the Switch-Feature data. Underlining is the best.

| | $\alpha = 0.1$ | | $\alpha = 0.5$ | | $\alpha = 1.0$ | | $\alpha = 2.0$ | | $\alpha = 5.0$ | |
| --- | --- | --- | --- | --- | --- | --- | --- | --- | --- | --- |
| | AUP | AUR | AUP | AUR | AUP | AUR | AUP | AUR | AUP | AUR |
| $\beta = 0.1$ | $0.53_{\pm 0.05}$ | $0.28_{\pm 0.18}$ | $0.26_{\pm 0.07}$ | $0.01_{\pm 0.00}$ | $0.18_{\pm 0.07}$ | $0.01_{\pm 0.00}$ | $0.12_{\pm 0.05}$ | $0.01_{\pm 0.00}$ | $0.14_{\pm 0.06}$ | $0.01_{\pm 0.00}$ |
| $\beta = 0.5$ | $0.56_{\pm 0.03}$ | $0.97_{\pm 0.01}$ | $0.91_{\pm 0.06}$ | $0.44_{\pm 0.28}$ | $0.52_{\pm 0.20}$ | $0.02_{\pm 0.01}$ | $0.19_{\pm 0.05}$ | $0.02_{\pm 0.00}$ | $0.16_{\pm 0.09}$ | $0.01_{\pm 0.00}$ |
| $\beta = 1.0$ | $0.55_{\pm 0.02}$ | $0.97_{\pm 0.01}$ | $0.89_{\pm 0.02}$ | $0.87_{\pm 0.02}$ | $0.98_{\pm 0.01}$ | $0.56_{\pm 0.10}$ | $0.71_{\pm 0.27}$ | $0.09_{\pm 0.09}$ | $0.28_{\pm 0.12}$ | $0.02_{\pm 0.00}$ |
| $\beta = 2.0$ | $0.54_{\pm 0.02}$ | $0.97_{\pm 0.01}$ | $0.86_{\pm 0.02}$ | $0.89_{\pm 0.02}$ | $\underline{0.98}_{\pm 0.01}$ | $\underline{0.80}_{\pm 0.03}$ | $0.99_{\pm 0.00}$ | $0.68_{\pm 0.06}$ | $0.50_{\pm 0.32}$ | $0.05_{\pm 0.07}$ |
| $\beta = 5.0$ | $0.54_{\pm 0.02}$ | $0.97_{\pm 0.01}$ | $0.87_{\pm 0.02}$ | $0.89_{\pm 0.02}$ | $0.97_{\pm 0.01}$ | $0.80_{\pm 0.03}$ | $0.99_{\pm 0.00}$ | $0.69_{\pm 0.05}$ | $0.99_{\pm 0.00}$ | $0.37_{\pm 0.09}$ |

Table 10: Effects of $\alpha$ and $\beta$ on the State data. Underlining is the best.

| | $\alpha = 0.1$ | | $\alpha = 0.5$ | | $\alpha = 1.0$ | | $\alpha = 2.0$ | | $\alpha = 5.0$ | |
| --- | --- | --- | --- | --- | --- | --- | --- | --- | --- | --- |
| | AUP | AUR | AUP | AUR | AUP | AUR | AUP | AUR | AUP | AUR |
| $\beta = 0.1$ | $0.54_{\pm 0.01}$ | $0.99_{\pm 0.00}$ | $0.67_{\pm 0.03}$ | $0.79_{\pm 0.05}$ | $0.69_{\pm 0.05}$ | $0.01_{\pm 0.00}$ | $0.32_{\pm 0.14}$ | $0.01_{\pm 0.00}$ | $0.53_{\pm 0.08}$ | $0.01_{\pm 0.00}$ |
| $\beta = 0.5$ | $0.52_{\pm 0.01}$ | $0.96_{\pm 0.00}$ | $0.66_{\pm 0.01}$ | $0.90_{\pm 0.01}$ | $0.77_{\pm 0.02}$ | $0.85_{\pm 0.01}$ | $0.88_{\pm 0.03}$ | $0.79_{\pm 0.03}$ | $0.77_{\pm 0.11}$ | $0.08_{\pm 0.14}$ |
| $\beta = 1.0$ | $0.52_{\pm 0.02}$ | $0.96_{\pm 0.00}$ | $0.66_{\pm 0.01}$ | $0.91_{\pm 0.00}$ | $0.77_{\pm 0.03}$ | $0.87_{\pm 0.01}$ | $\underline{0.90}_{\pm 0.02}$ | $\underline{0.82}_{\pm 0.01}$ | $0.88_{\pm 0.09}$ | $0.23_{\pm 0.29}$ |
| $\beta = 2.0$ | $0.52_{\pm 0.01}$ | $0.96_{\pm 0.00}$ | $0.65_{\pm 0.02}$ | $0.92_{\pm 0.00}$ | $0.77_{\pm 0.02}$ | $0.88_{\pm 0.01}$ | $0.89_{\pm 0.02}$ | $0.82_{\pm 0.01}$ | $0.97_{\pm 0.01}$ | $0.70_{\pm 0.01}$ |
| $\beta = 5.0$ | $0.52_{\pm 0.01}$ | $0.96_{\pm 0.00}$ | $0.65_{\pm 0.01}$ | $0.91_{\pm 0.00}$ | $0.76_{\pm 0.02}$ | $0.88_{\pm 0.01}$ | $0.89_{\pm 0.03}$ | $0.82_{\pm 0.01}$ | $0.97_{\pm 0.01}$ | $0.70_{\pm 0.02}$ |

## F DISTRIBUTION ANALYSIS OF PERTURBATIONS

To investigate whether the perturbed samples are within the original dataset's distribution, we first compute the distribution of the original samples by kernel density estimation[4] (KDE). Subsequently,

---

[4] https://scikit-learn.org/stable/modules/generated/sklearn.neighbors.KernelDensity.html

Table 11: Effects of $\alpha$ and $\beta$ on MIMIC-III mortality. We mask 20% data and replace the masked data with the overall average over time for each feature. Underlining is the best.

| | $\alpha = 0.001$ | | $\alpha = 0.005$ | | $\alpha = 0.01$ | | $\alpha = 0.1$ | | $\alpha = 1.0$ | |
|---|---|---|---|---|---|---|---|---|---|---|
| | Acc | CE | Acc | CE | Acc | CE | Acc | CE | Acc | CE |
| $\beta = 0.001$ | $0.982_{\pm 0.003}$ | $0.124_{\pm 0.007}$ | $0.983_{\pm 0.003}$ | $0.122_{\pm 0.007}$ | $0.984_{\pm 0.002}$ | $0.120_{\pm 0.006}$ | $0.993_{\pm 0.001}$ | $0.094_{\pm 0.004}$ | $0.997_{\pm 0.001}$ | $0.087_{\pm 0.004}$ |
| $\beta = 0.005$ | $0.981_{\pm 0.002}$ | $0.123_{\pm 0.007}$ | $0.984_{\pm 0.002}$ | $0.123_{\pm 0.006}$ | $0.984_{\pm 0.003}$ | $0.121_{\pm 0.007}$ | $0.993_{\pm 0.002}$ | $0.095_{\pm 0.006}$ | $0.996_{\pm 0.001}$ | $0.087_{\pm 0.005}$ |
| $\beta = 0.01$ | $0.980_{\pm 0.003}$ | $0.124_{\pm 0.007}$ | $\underline{0.980}_{\pm 0.002}$ | $0.127_{\pm 0.007}$ | $0.984_{\pm 0.002}$ | $0.121_{\pm 0.007}$ | $0.994_{\pm 0.002}$ | $0.094_{\pm 0.004}$ | $0.996_{\pm 0.001}$ | $0.087_{\pm 0.004}$ |
| $\beta = 0.1$ | $0.980_{\pm 0.002}$ | $0.127_{\pm 0.007}$ | $\underline{0.980}_{\pm 0.003}$ | $0.127_{\pm 0.007}$ | $0.983_{\pm 0.003}$ | $0.123_{\pm 0.007}$ | $0.992_{\pm 0.002}$ | $0.098_{\pm 0.006}$ | $0.997_{\pm 0.001}$ | $0.087_{\pm 0.005}$ |
| $\beta = 1.0$ | $0.981_{\pm 0.002}$ | $0.127_{\pm 0.006}$ | $0.981_{\pm 0.003}$ | $0.128_{\pm 0.008}$ | $0.983_{\pm 0.002}$ | $0.123_{\pm 0.007}$ | $0.989_{\pm 0.002}$ | $0.106_{\pm 0.007}$ | $0.996_{\pm 0.001}$ | $0.088_{\pm 0.005}$ |

Table 12: Difference between the distribution of different perturbations and the original distribution.

| | RARE-TIME | | RARE-OBSERVATION | |
|---|---|---|---|---|
| PERTURBATION TYPE | KDE-SCORE ↑ | KL-DIVERGENCE ↓ | KDE-SCORE ↑ | KL-DIVERGENCE ↓ |
| ZERO PERTURBATION | $-25.242$ | $0.0523$ | $-23.377$ | $0.0421$ |
| MEAN PERTURBATION | $-30.805$ | $0.0731$ | $-26.421$ | $0.0589$ |
| EXTRMASK PERTURBATION | $-22.532$ | $0.0219$ | $-19.102$ | $0.0104$ |
| CONTRALSP PERTURBATION | $-23.290$ | $0.0393$ | $-22.732$ | $0.0386$ |

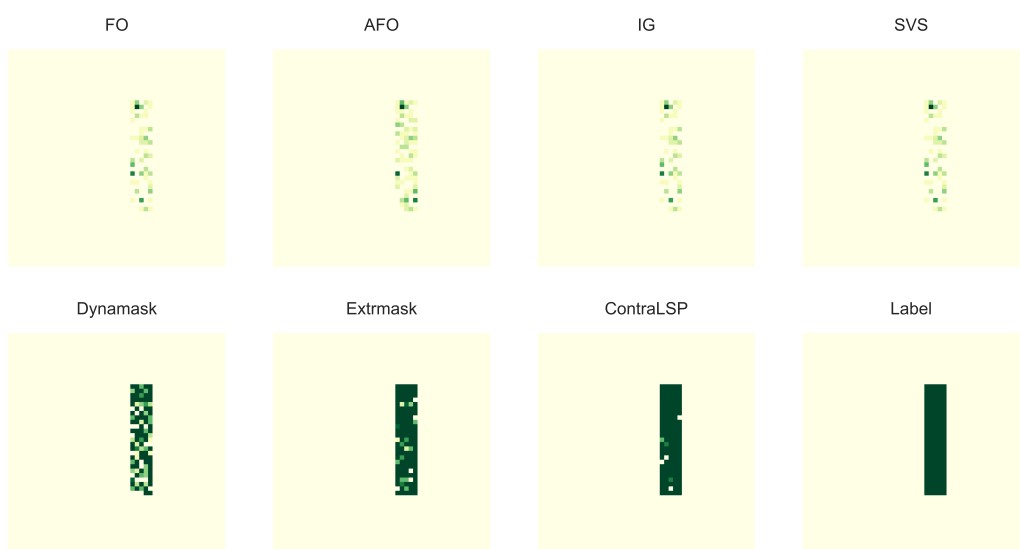

Figure 8: Saliency maps produced by various methods for Rare-Time experiment.

we assess the log-likelihood of each perturbed sample under the original distribution, called as KDE-score, where closer to $0$ indicates a higher likelihood of perturbed samples originating from the original distribution. Additionally, we quantify the KL-divergence between the distribution of perturbed samples and original samples, where a smaller KL means that the two distributions are closer. We conduct experiments on the Rare-Time and Rare-Observation datasets and the results are shown in Table 12. It demonstrates that our ContraLSP's perturbation is more akin to the original distribution compared to the zero and mean perturbation. Furthermore, Extrmask performs best because it generates perturbations only from current samples, and therefore the generated perturbations are not guaranteed to be uninformative. This conclusion aligns with the visualization depicted in Figure 1.

## G  ILLUSTRATIONS OF SALIENCY MAPS

Saliency maps represent a valuable technique for visualizing the significance of features, and previous works (Alqaraawi et al., 2020; Tonekaboni et al., 2020; Leung et al., 2023), particularly in multivariate time series analysis, have demonstrated their utility in enhancing the interpretative aspects of the results. We also demonstrate the saliency maps of the benchmarks and our method for each dataset: (i) the saliency maps for the rare experiments are shown in Figure 8, 9, 11, and 10, (ii)

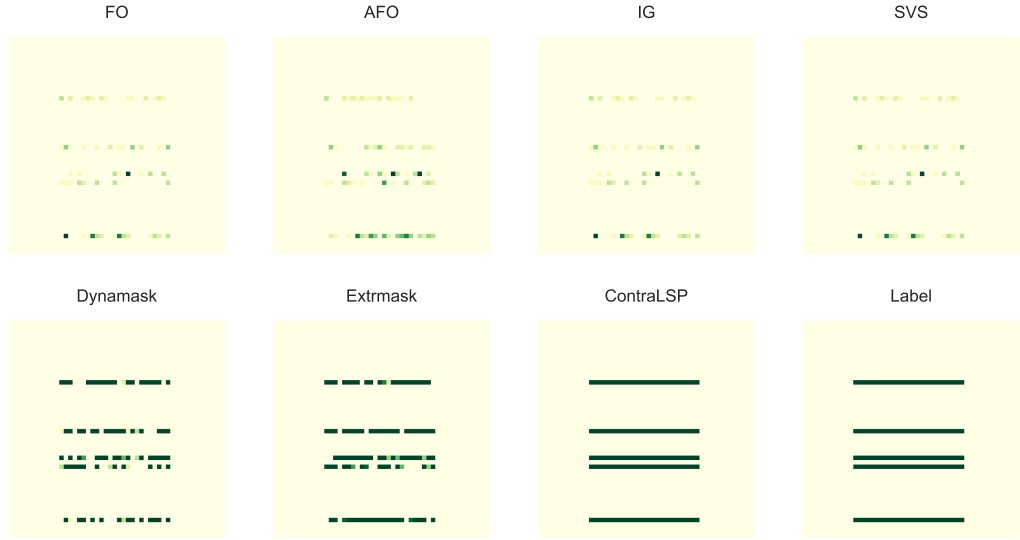

Figure 9: Saliency maps produced by various methods for Rare-Observation experiment.

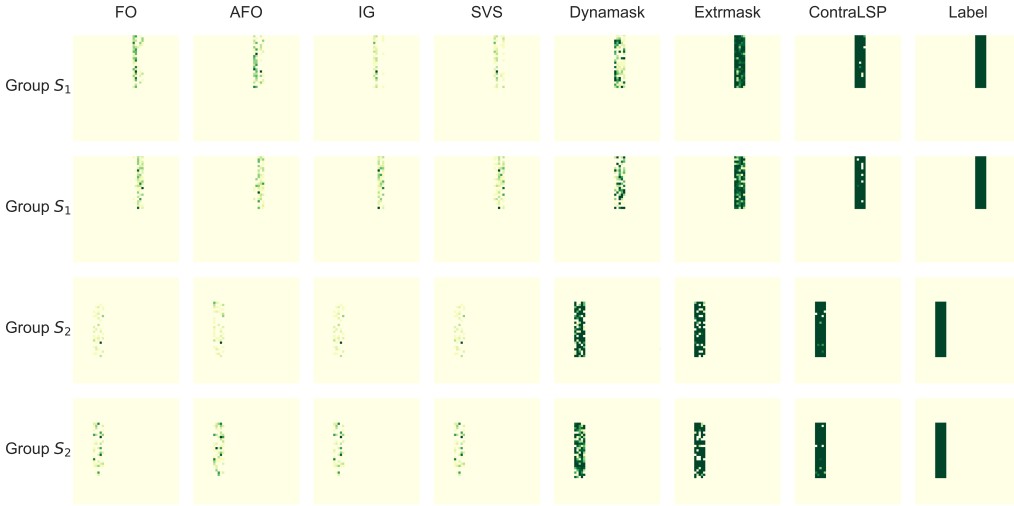

Figure 10: Saliency maps produced by various methods for Rare-Time (Diffgroups) experiment.

the Switch-Feature and State saliency maps are shown in Figure 12 and Figure 13, respectively, (iii) and the saliency maps for the MIMIC-III mortality are in Figure 14.

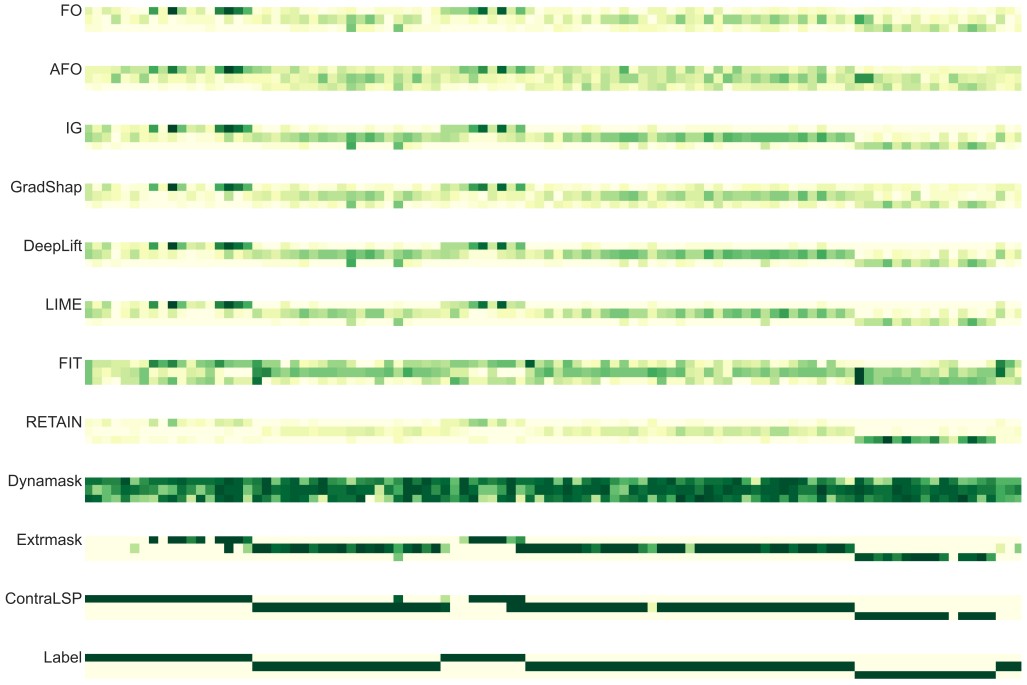

Figure 11: Saliency maps produced by various methods for Rare-Observation (Diffgroups) experiment.

Figure 12: Saliency maps produced by various methods for Switch-Feature data.

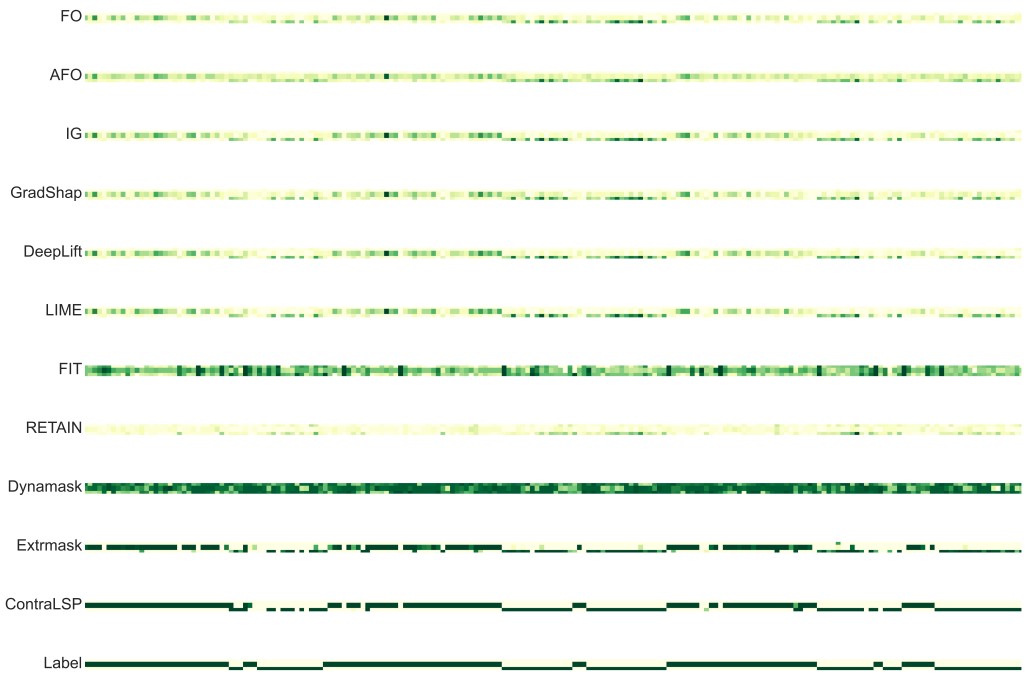

Figure 13: Saliency maps produced by various methods for State data.

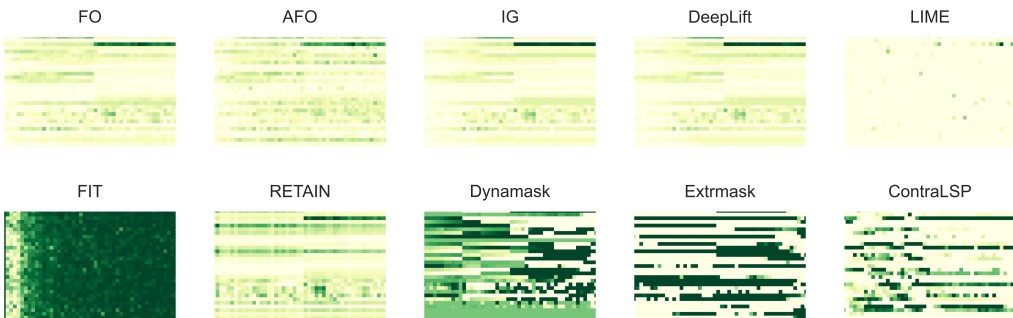

Figure 14: Saliency maps produced by various methods for MIMIC-III Mortality data.

