# OpenReview forum: "Explaining Time Series via Contrastive and Locally Sparse Perturbations"
_ICLR.cc/2024/Conference — ICLR 2024 poster_

### Official Review · Reviewer_MZY5 · 2023-10-24

**Soundness:** 3 good
**Presentation:** 3 good
**Contribution:** 3 good
**Rating:** 6
**Confidence:** 2

**Summary:**

-	The paper proposes a Contrastive and Locally Sparse Perturbations (ContraLSP) framework, which utilizes contrastive learning techniques to render non-salient features uninformative during training. The sparse gate with $\ell_0$ regularization can aid in feature selection. The proposed method exhibits strength in both with-box and black-box scenarios.

**Strengths:**

-	The paper demonstrates its novelty by incorporating contrastive learning techniques into explainable time series tasks. Contrastive learning is a suitable solution for distinguishing informative and non-informative components.
-	The paper is well-written and easy to read, and the figures effectively aid in comprehending the main ideas.
-	Through the use of perturbation methods, ContraLSP remains relatively unaffected by noise and uninformative parts.
-	The paper shows the performance enhancement of ContraLSP across a wide range of datasets, surpassing existing methods. The authors cover various tasks in Rare-Time, Rare-Observation, and various real-world datasets.

**Weaknesses:**

-	Please refer to questions.

**Questions:**

I will happily raise the score if the authors can address the following questions:

-	1. Although the authors discuss the selection of positive and negative samples in Appendix B, the selection of positive and negative pairs in time series is quite controversial because the proximity of data samples does not guarantee similarities [1,2,…]. The method in Appendix B appears too naive and may pose a risk of incorrect sampling for time series pair selection.
-	2. Can you provide a more specific explanation of why the counterfactual of non-salient features is superior to ignoring that part (e.g., setting it to zero), as shown in Figure 1? Even though a zero value of $x$ does not affect the training to minimize the loss of prediction with weight $w$ as $wx$, using counterfactuals can have adverse effects.
-	3. The learned mask in Figure 4 appears to exhibit similar behavior to a hard mask rather than other smooth masks. Can you clarify how the learned function $\tau(\cdot)$ behaves in a multi-dimensional context? I have read the ablation study in Table 3.
-	4. What is the difference in using the $\ell_0$ norm in Section 4.2 of your methods compared to previous methods that use the $\ell_0$ norm?

[1] Unsupervised Representation Learning for Time Series with Temporal Neighborhood Coding, ICLR 2021.
[2] TS2Vec: Towards Universal Representation of Time Series, AAAI 2022.

**Details Of Ethics Concerns:**

No ethics concerns.

---

> ### Author Response · Authors · 2023-11-19
> **Response to Reviewer MZY5 (part 1/2)**
>
> Dear Reviewer,
>
> We thank the reviewer for the detailed constructive feedback on our work and answer the questions below:
>
> 1. **The selection of positive and negative pairs in time series is too naive in Appendix B.**
>
>    **Response**: Thanks for your insights about the nuanced aspect of contrastive sample selection! The concept and strictness of data samples' proximity depend on the specific method used. Specifically, as you mentioned, proximal data means subseries within the stationary window in [1], and augmented subseries in [2]. Both of these are within the same instance and are relatively weakly constrained in terms of similarity. In contrast, our sampling employs an instance-wise comparison using a more stringent concept for similarity (Manhattan distance), and we have added performance comparisons of different similarity metrics in the latest version (see Table 8 in Appendix E.1). Our stricter criteria for similarity likely reduce the risk of noisy sampling. The core rationale behind this is that our aim in choosing contrastive samples is to craft perturbations rather than to learn a generalizable representation, which means a straightforward method is already effective.
>
> [1] Tonekaboni, et al.  Unsupervised Representation Learning for Time Series with Temporal Neighborhood Coding. In *ICLR*, 2021.
>
> [2] Yue, et al.  TS2Vec: Towards Universal Representation of Time Series. In *AAAI*, 2022.
>
>
> 2. **Why the counterfactual of non-salient features is superior to ignoring that part.**
>
>    **Response**: Thanks for your constructive comment. If it is an important feature, our goal is to have the counterfactual produce the opposite conclusion so that the loss $\mathcal{L}(f(x), f(x, m))$ increases, which leads to $m$ optimizing to $1$ and then minimizing the loss. For example, a regression task $y = f(x)$ where we need to perturb whether an observation $x=1$ is important or not. During training, our perturbation $x^r$ learns to a counterfactual $x^r=-3$ and a perturbation of other method is $x^{r \prime}=0$. If it is not salient, the change in $x$ is independent of $y$, so that $f(1)\approx f(-3)\approx f(0)$ and $m=0$. But if it is salient, $\mathcal{L}(f(1), f(-3)) > \mathcal{L}(f(1), f(0)) $ means that our $m$ is more likely to learn $1$, i.e, the loss changes from $\mathcal{L}(f(1), f(-3))$ to $\mathcal{L}(f(1), f(1))$.
>
>    Moreover, we posit that perturbations in the function $f$ are not universally amenable to being substituted with zero; such substitutions may result in out of distribution outcomes within the explained model. Figure 1 illustrates the absence of zeros in the data domain during the intermediate period of time, while the $0$ and mean perturbations shift in shape. In instances where a trained model is confronted with previously unseen data, deleterious effects may occur [3], leading to erroneous judgments. Note that the counterfactual perturbation described in Eq. (2) is learned while concurrently constraining its extent using an $l_1$ norm term. This constraint serves to mitigate the generation of excessively anomalous counterfactual instances.
>
>
> [3] Shen, et al. Towards out-of-distribution generalization: A survey. *arXiv:2108.13624*, 2021.
>
> 3. **The learned mask in Figure 4 appears to exhibit similar behavior to a hard mask rather than other smooth masks. Can you clarify how the learned function $\tau()$ behaves in a multi-dimensional context?**
>
>    **Response**: Thanks for the nice question. It is imperative that masks adopt a hard nature (e.g., Bernoulli distribution) rather than being continuous [4]. While Dynamask and Extrmask (cited in the paper) have explored masking techniques using continuous masks and using $l_1$ regularization for constraints $m$. It involves a continuous deformation of the input towards a baseline value. Consider a toy example, assuming that masks are continuous and $x_1>x_2$. It is possible that $m_1<m_2$ but $x_1m_1>x_2m_2$  for the unsalient feature $x_2$. Therefore, we use a binary-skewed mask, which is constrained by the $l_0$-like regularization and can be close to the hard mask (as in Fig. 2). Moreover, the sigmoid-weighted unit with the temporal trend to smooth this hard mask by penalizing irregular shapes ($A(m)$ in Eq.(1)).
>
>    In the trend function $\tau(\cdot)$, we employ an independent MLP for each observation $x[:, d]$ to find its trend, whose details are shown in Table 7 (see Appendix D.4). Consequently, the multidimensional contexts remain mutually independent. The function $\tau(\cdot)$ is designed to exclusively capture a singular temporal dimension associated with the trend, utilized for the purpose of smoothing. We have modified the description of Fig. 2 in the latest version for clarity.
>
>
> [4] Queen, et al. Encoding Time-Series Explanations through Self-Supervised Model Behavior Consistency. In *NeurIPS*, 2023.

---

> ### Author Response · Authors · 2023-11-19
> **Response to Reviewer MZY5 (part 2/2)**
>
> 4. **What is the difference in using the l0 norm in Section 4.2 of your methods compared to previous methods that use the l0 norm?**
>
>    **Response**: Thanks for your constructive comment. ContraLSP distinguishes itself significantly from previous methods in that we do not use sparsity constraints such as the $l_1$ norm (Dynamask and Extrmask), $l_0$ norm [5, 6], $l_2$ norm [7] to select salient sub-features. Our mask is just skewed binary but not purely containing 0 and 1 due to the smooth constraint. We apply an $l_0$-like constraint, i.e., by injecting random noise into $\mu^\prime $ and representing the $l_0$ norm with a Gaussian error function. The expected regularization term is simply the sum of the probabilities (in Appendix A) that sparse gates are active or $\sum_{t,d}P(\mu^\prime[t,d]+\epsilon[t, d] > 0)$. This sum is equal to $\sum_{t,d}\Psi \left(\frac{\mu^\prime[t,d]}{\delta }\right)$, where $\Psi $ is the standard Gaussian CDF. The random noise $\epsilon$ of the model plays two important roles. Firstly, facilitates the training of weights in a binary model, specifically the gating network in Eq. (3). And secondly, it enables the re-evaluation of features that undergo sparsification during the early stages of training [8].
>
> [5] Bastings, et al. Interpretable Neural Predictions with Differentiable Binary Variables. In *ACL*, 2019.
>
> [6] Schlichtkrull, et al. Interpreting graph neural networks for nlp with differentiable edge masking. In *ICLR*, 2021.
>
> [7] Yu, et al. Graph information bottleneck for subgraph recognition. In *ICLR*, 2022.
>
> [8] Yang, et al. Locally sparse neural networks for tabular biomedical data. In *ICML*, 2022.

---

> ### Author Response · Authors · 2023-11-21
> **Response to Reviewer MZY5 before the end of discussion**
>
> Dear Reviewer MZY5,
>
> Since the End of author/reviewer discussions is just in one day, may we know if our response addresses your main concerns? If so, we kindly ask for your reconsideration of the score.
>
> Should you have any further advice on the paper and/or our rebuttal, please let us know and we will be more than happy to engage in more discussion and paper improvements. We would really appreciate it if our next round of communication could leave time for us to resolve any of your remaining or new questions.
>
> Thank you so much for devoting time to improving our paper!

---

> > ### Comment · Reviewer_MZY5 · 2023-11-22
> > **Rebuttal follow-up**
> >
> > I first argue that I am not an expert in this particular direction, so I am not too certain how significant these counterfactual perturbations (this is the reason for Confidence: 2). Thus, I have awaited the comments of other reviewers.
> >
> > I have carefully reviewed the authors' rebuttal and the feedback from other reviewers. I thank for the authors for their comprehensive response and the additional effort they have put into enhancing the manuscript.
> > I think the all of my concerns are well-discussed due to the powerful rebuttal of the authors.
> > Therefore, I will raise the score for the manuscript (5->6).

---

### Official Review · Reviewer_9mjJ · 2023-10-31

**Soundness:** 3 good
**Presentation:** 2 fair
**Contribution:** 2 fair
**Rating:** 5
**Confidence:** 3

**Summary:**

This manuscript presents a new method in the explainability of time series predictions. The task here is given a multivariate time series of factors to predicting an output variable to identify the regions of the input that are most predictive of the output, here defined by a binary mask. The presented technique uses a perturbative approach to compute the binary mask on the input factors. While perturbative approaches have been considered, it differs from other approaches in how it produces these perturbations. They use a contrastive (triplet) loss across samples  a smoothed sparse gate. They provide a series of experiments comparing performance to other perturbative and other approaches, such as Dynamask and Shapley Features. They use a synthetic white-box experiment with rare observation or time salience and show improved recall with their method compared to others, although all methods showed high precision. Similarly on a synthetic state-switching task they find improved recall, and perform ablations to show how  different inputs vary. They provide further examples using classification and with a real-world mortality task. There is a comprehensive supplement giving further documentation for the methods and experiments.

**Strengths:**

* The approach is original and I believe a good contribution to existing methodologies.

* They present a comprehensive set of experiments that are well motivated and the result achieves SOTA performance on many of them.

* While the writing and motivation is often unclear, the math behind the method is very clearly explained.

**Weaknesses:**

* The advance feels somewhat incremental and the experiments performed are near-identical to the Dynamask paper (cited in text).

* It was unclear to me how hyperparameters were selected for each experiment. This is important as changing the expected sparsity could have a dramatic effect on recall performance. More generally there are now a family of approaches for interpretability and it is not clear what the respective strengths and weaknesses are of each. This manuscript suggests their method is superior to all others, but a discussion of which types of data each method is suited to would be helpful.

* The manuscript is hard to follow the text as the writing and motivation is not clear in a number of points. Terms are not always introduced in order and it is hard to appreciate the innovations is. One of my reservations about this manuscript is even if the algorithm is novel it will be hard for others to appreciate.

* In the white box experiments I had trouble appreciating the experimental design, which made it difficult to evaluate. Moreover the largest difference between methods was in the information-based metrics, which seemed to scale quite nonlinearly with recall  (fine tuning/hyperparameters).

**Questions:**

* The authors mention treatment of inter-sample correlations as an important component of the technique, but I do not see clear evidence of this.

* Can you explain what in-domain and negative samples refer to “Other perturbations could be either not uninformative or not in-domain, while ours is counterfactual that is toward the distribution of negative samples “

* “ To cope with it, locally stochastic gates Yang et al. (2022) consider an instance-wise selector that employs heterogeneous samples. Lee et al. (2022) takes a self-supervised way with unlabeled samples to enhance stochastic gates that encourage the model explainability meanwhile.” The terms introduced here (stochastic gates, heterogeneous samples) are not defined. The writing is unclear as well.

* I found the description of the ‘Datasets and Benchmarks’ in 5.1 WHITE-BOX REGRESSION SIMULATION very unclear, making it hard to follow the experiments.

* “our method significantly outperforms all other benchmarks.” I do not see any tests of significance.

* Figure 5 I found unclear.

---

> ### Author Response · Authors · 2023-11-19
> **Response to Reviewer 9mjJ (part 1/4)**
>
> Dear Reviewer,
>
> We greatly appreciate your insightful comments! Here are our responses to the comments.
>
> 1. **The advance feels somewhat incremental and the experiments performed are near-identical to the Dynamask paper (cited in text).**
>
>    **Response**: We believe our approach significantly differs from the Dynamask method proposed in Crabbe et al. (2021), both in an important conceptual way, as well as in the details of the formulation, beyond the experimental datasets and metrics.
>
>    The key conceptual difference is that Dynamask adapts image perturbations [1] to multivariate time series, where the perturbation operator $\Phi (x, m)$ accounts for this temporal information. Specifically, a perturbation is constructed for each instance, utilizing the feature values at adjacent time steps. This allows Dynamask to build meaningful perturbation. In contrast, our ContraLSP involves optimizing two components: the mask and the perturbation. It employs negative samples for constructing counterfactual perturbations by contrastive learning. Our masks integrate sample-specific sparse gates with the smooth constraint, producing binary-skewed masks on time series data.
>
>    In terms of the formulation details, we can simply express the mask optimization of Dynamask as: $\mathcal{R}(x_i, m_i) = \alpha_1 \sum ||\text{vecsort}(m_i)-r_a|| + \alpha_2\sum ||m_{t+1,i}-m_{t,i}||$, where the second term is the smoothing of masks. Dynamask uses area constraints in the first term, but it is not clear how to choose the hyperparameter $a$ on complex data. Differently, the mask optimization of ContraLSP is expressed as an $l_0$-like regularization as: $\mathcal{R}(x_i, m_i) = \alpha \sum_{t=1}^T\sum_{d=1}^D\left(\frac{1}{2}+\frac{1}{2} \text{erf}\left(\frac{\mu^\prime _i[t,d]}{\sqrt{2} \delta }\right)\right)$, where the smooth vectors $\mu^\prime_i$ restrict the penalty for jump saliency over time. By injecting a random noise into $\mu^\prime $ and representing the $l_0$ norm with a Gaussian error function (see Appendix A), it can facilitate the training of weights in a binary model, specifically the sparse gating network. Furthermore, we optimize learned counterfactuals in Eq. (2) instead of using the meaningful perturbation (using temporal Gaussian blur does not support optimization). It provides more uninformative information and thus improves explanation accuracy.
>
>    The experimental datasets and metrics are consistent with Dynamask for fair comparisons, and most time series explanation methods use similar experiments. In addition, when explaining the regression model, we added different groups of rare experiments to validate the effect of local masks on the inter-sample correlation, which further distinguishes the other methods.
>
>
> [1] Fong, et al. Interpretable explanations of black boxes by meaningful perturbation. In *CVPR*, 2017.

---

> ### Author Response · Authors · 2023-11-19
> **Response to Reviewer 9mjJ (part 2/4)**
>
> 2. **How hyperparameters were selected for each experiment?  The method is superior to all others, but a discussion of which types of data each method is suited to would be helpful.**
>
>    **Response**: Thanks for the nice question. Firstly, it is imperative to clarify that determining the proportion of salient features in an exact model remains elusive. Therefore the determination of hyperparameters across distinct datasets proves challenging [2, 3] and we do not focus on it in this manuscript. In fact, we have given the specific parameters chosen for each dataset (see Table 6 in Appendix D.4). Specifically, the four Rate experiments used a default value of 0.1 (both $\alpha$ and $\beta$) determined by experience and other mask-based methods. Subsequently, we conducted ablation experiments (refer to Tables 9 and 10 in Appendix E.2) centered around these default values to identify optimal parameters for the Switch-Feature and State datasets. It is noteworthy that selecting a subset of features proves to be unstable for most interpretation methods when determining the regularization hyperparameter. We mentioned that it's in the limit: "However, an inherent limitation of our method is the selection of sparse parameters, especially when dealing with different datasets...". For the MIMIC-III dataset, we add the same ablation experiments in the latest version. We employ Accuracy and Cross-Entropy as metrics and show the average substitution in the following table and select $\alpha=0.005$ and $\beta=0.01$ as deterministic parameters on the MIMIC-III dataset. In the results, our method demonstrates suitability and superiority over other baseline methods across both regression and classification tasks, while some baselines (e.g., DeepLIFT and FIT) are not applicable to evaluate white-box regression models (see Section 5.1).
>
> [1] Bansal, et al. Sam: The sensitivity of attribution methods to hyperparameters. In *CVPR*, 2020.
> [2] Agarwal, et al. Explaining image classifiers by removing input features using generative models.  In *ACCV*, 2020.
>
> Table 11: Effects of $\alpha$ and $\beta$  on MIMIC-III mortality. We mask 20\% data and replace the masked data with the overall average over time for each feature. **bold** is the best and values are reported as Accuracy - Cross-Entropy.
>
> |                |              $\alpha$= 0.001              |              $\alpha$= 0.005              |               $\alpha$= 0.01               |               $\alpha$= 0.1                |               $\alpha$= 1.0               |
> | :------------: | :---------------------------------------: | :---------------------------------------: | :----------------------------------------: | :----------------------------------------: | :---------------------------------------: |
> | $\beta$= 0.001 | ${0.982}{\pm0.003}$ - ${0.124}{\pm0.007}$ | ${0.983}{\pm0.003}$ - ${0.122}{\pm0.007}$ | ${0.984}{\pm0.002}$ - ${0.120}{\pm0.006}$  | ${0.993}{\pm0.001}$ - ${0.094}{\pm0.004}$  | ${0.997}{\pm0.001}$ - ${0.087}{\pm0.004}$ |
> | $\beta$= 0.005 | ${0.981}{\pm0.002}$ - ${0.123}{\pm0.007}$ | ${0.984}{\pm0.002}$ - ${0.123}{\pm0.006}$ | ${0.984}{\pm0.003}$ - ${0.121}{\pm0.007}$  | ${0.993}{\pm0.002}$ - ${0.095}{\pm0.006}$  | ${0.996}{\pm0.001}$ - ${0.087}{\pm0.005}$ |
> | $\beta$= 0.01  | ${0.980}{\pm0.003}$ - ${0.124}{\pm0.007}$ |  **0.980$\pm$0.002 -  0.127$\pm$0.007**   | ${0.984}{\pm0.002}$ - ${0.121}{\pm0.007}$  | ${0.994}{\pm0.002}$ - ${0.094}{\pm0.004}$  | ${0.996}{\pm0.001}$ - ${0.087}{\pm0.004}$ |
> |  $\beta$= 0.1  | ${0.980}{\pm0.003}$ - ${0.127}{\pm0.007}$ | ${0.980}{\pm0.003}$ - ${0.127}{\pm0.007}$ | ${0.983}{\pm 0.003}$ - ${0.123}{\pm0.007}$ | ${0.992}{\pm0.002}$ - ${0.098}{\pm 0.006}$ | ${0.997}{\pm0.001}$ - ${0.087}{\pm0.005}$ |
> |  $\beta$= 1.0  | ${0.981}{\pm0.002}$ - ${0.127}{\pm0.006}$ | ${0.981}{\pm0.003}$ - ${0.128}{\pm0.008}$ | ${0.983}{\pm0.002}$ - ${0.123}{\pm0.007}$  | ${0.989}{\pm0.002}$ - ${0.106}{\pm0.007}$  | ${0.996}{\pm0.001}$ - ${0.088}{\pm0.005}$ |

---

> ### Author Response · Authors · 2023-11-19
> **Response to Reviewer 9mjJ (part 3/4)**
>
> 3. **The manuscript is hard to follow the text as the writing and motivation is not clear in a number of points.**
>
>    **Response**: Thanks for pointing this question out, we acknowledge the concerns raised regarding clarity and motivation. We are dedicated to revising the manuscript to enhance the organization, ensuring that terms are introduced in a logical order for better comprehension. The latest version has corrected inappropriate sentences, and a comprehensive review will be conducted for the final version.
>
>
> 4. **The design and description of the white box experiments lack clarity (Both W4 and Q4). Moreover the information-based metrics seem to scale quite nonlinearly with recall.**
>
>    **Response**: Thanks for your constructive comment. We apologize for the lack of clarity in the description of the white box experiments and have modified the corresponding section in the latest version. In fact, we follow the experimental setup of Crabbe et al (2021), using sparse white-box regressors whose predictions depend only on the known sub-features. As discussed in the challenge, under-considering the inter-correlation of samples (heterogeneous samples) would result in significant generalization errors, which can degrade performance. We also consider sample-specific explanations of the predicted results, thus designing the different groups for regressors. The results in Table 1 show that the performance of mask-based methods at the baseline significantly deteriorates, while ContraLSP remains relatively unaffected.
>
>    The information-based metrics are introduced by Crabbe et al (2021): the information $I_m (a)= - \sum_{[i,t,d]\in a} \ln(1 - m_i[t,d])$ and mask entropy $S_m (a)= - \sum_{[i,t,d]\in a} m_i[t,d]\ln(m_i[t,d])+ (1-m_i[t,d])\ln(1-m_i[t,d])$,  where $a$ represents true salient features. They serve as tools to illustrate how metric information is not linearly correlated with the recall rate. These metrics aim to encourage a nearly binary form of the mask within mask-based methods to enhance simplicity. Specifically, this entails promoting a distinct contrast between low and high saliency scores. Moreover, to make the notion of legibility quantitative, those two metrics use a parallel with Shannon's theory.
>
> 5. **How ContraLSP handles inter-sample correlation.**
>
>    **Response**: Thank you for the thoughtful comment. We want to be clear that ContraLSP is not handling inter-sample correlation, and we don't mention that. The inter-sample correlation of features often leads to notable generalization errors. To cope with it, we employ local sparse gates (as described in Eq. (3)) allowing samples to be independent of each other.  In addition, our investigation performs counterfactual perturbations using contrastive learning through inter-sample variation, which goes beyond the instance-level saliency methods by focusing on understanding both the overall and specific model's behavior across groups. Therefore, we designed different groups for white-box regressors to study it.
>
> 6. **Can you explain what in-domain and negative samples in Figure 1?**
>
>    **Response**: Thanks for the nice question. As shown in Figure 1, the entire data domain encompasses two distinct sample categories delineated by the red and blue distributions. Perturbation of the original features (the red line) is necessary to find the salient sub-features. Some perturbation methods, such as mean perturbation (the blue line) and 0-value perturbation (the orange line), are not in the data domain for an intermediate period of time, thus we call it not "in-domain". In contrast, our perturbations learn counterfactual features (the black line) that categorize its differences from the original sample. We call the blue distribution the distribution of negative samples that are not consistent with the classification results of the original features. Consequently, our perturbations gravitate towards the distribution of negative samples through contrastive learning.

---

> ### Author Response · Authors · 2023-11-19
> **Response to Reviewer 9mjJ (part 4/4)**
>
> 7. **The terms introduced here (stochastic gates, heterogeneous samples) are not defined.**
>
>    **Response**: Thank you for your careful comment. The "local stochastic gates" is a gating structure used by Yang et al. [3] and originally proposed by Yamada et al. [4]. It injects a random noise into the mask and is used for sparsified feature selection. The mathematical form (see Eq. (3) in section 4.2) can be described as $m_i = \mu_i + \epsilon_i$, where $\epsilon \sim \mathcal{N}(0, \delta ^2)$. Heterogeneous samples occur when the samples have differences, where different black box models act on different samples thus eliciting differing salient features. We therefore designed different groups of experiments within the white-box regression simulation (see section 5.1) to validate the effectiveness of our method.
>
>    We recognize the lack of description in this sentence and modify it to "To cope with it, local stochastic gates~\citep{yang22i} consider an instance-wise selector to heterogeneous samples, accommodating cases where salient features vary among samples. \citet{lee2021self} takes a self-supervised way to enhance stochastic gates that encourage the model's sparse explainability meanwhile." in the latest version. Thanks again for your careful reading.
>
> [3] Yang, et al. Locally sparse neural networks for tabular biomedical data. In *ICML*, 2022.
>
> [4] Yamada, et al. Feature selection using stochastic gates. In *ICML*, 2020.
>
>
> 8. **“our method significantly outperforms all other benchmarks.” I do not see any tests of significance.**
>
>    **Response**: We evaluate the interpretability performance of various algorithms on four rare-experiments, presenting results in Table 1 in section 5.1. It shows that our method demonstrates significant outperformance across all metrics, except AUP, compared to other benchmarks. Notably, AUP is not effective as a performance discriminator in rare-experiment scenarios due to all algorithms being close to 1. Specifically, ContraLSP has an average 23% improvement in AUC, an average 49% improvement in Information, and an average 88% reduction in Entropy over the leading baseline across rare-experiments. It suggests that ContraLSP identifies really salient features in all experiments.
>
> 9. **Figure 5 I found unclear.**
>
>       **Response**: We appreciate the reviewer's feedback regarding the clarity of our illustrations, particularly Figure 5. We present a comparison between the perturbations generated by ContraLSP and Extrmask on the Rare-Observation (Diffgroups) experiment, as shown in Figure 5 (the updated version). The background color represents the mask value, with darker colors indicating higher values. We randomly select a sample in each of the two groups and sum all observations. ContraLSP provides counterfactual information, yet Extrmask's perturbation is close to 0. This suggests ContraLSP provides counterfactual perturbations that is more beneficial to non-salient areas while emphasizing the mask more distinctly in salient areas compared to Extrmask. This is consistent with the findings in Figure 1. Moreover, the performance of Extrmask is poor when samples exhibit heterogeneity among distinct groups, whereas ContraLSP remains relatively unaffected.
>
> Thank you again for your constructive comments and efforts to help improve our work.

---

> ### Author Response · Authors · 2023-11-22
> **Response to Reviewer 9mjJ (near the end of rebuttal)**
>
> Dear Reviewer 9mjJ,
>
> We tried our best to address your concerns. Since the End of author/reviewer rebuttal is coming soon, may we know if our response addresses your main concerns? If so, we kindly ask for your reconsideration of the score.
>
> Should you have any further advice on the paper and/or our rebuttal, please let us know and we will be more than happy to engage in more discussion and paper improvements.
>
> Thank you so much for devoting time to improving our work!

---

### Official Review · Reviewer_zXTo · 2023-10-31

**Soundness:** 3 good
**Presentation:** 3 good
**Contribution:** 3 good
**Rating:** 6
**Confidence:** 2

**Summary:**

This paper presents ContraLSP, a locally sparse model that introduces counterfactual samples to build uninformative perturbations but keeps distribution using contrastive learning.

Note that this paper does not quite match the expertise of my research and I have made the comments to AC.

**Strengths:**

1. The idea of using contrastive loss seems to be a new idea
2. In the experiment section, the authors provide a comprehensive comparison with multiple baseline models
3. The paper is well written.

**Weaknesses:**

I mainly have some questions to the author:
1. I curious about if the method is scalable to high dimension data. For example, video sequences?
2. What is the $\alpha$ and $\beta$ values you use for each dataset? and how do you determine their values?
3. In section 4.1, I wonder why you choose Manhattan distance rather than more conventional Euclidean distance? Optionally, other metrics like cosine distance might work better for contrastive learning?

**Questions:**

see above

---

> ### Author Response · Authors · 2023-11-19
> **Response to Reviewer zXTo (part 1/2)**
>
> Dear Reviewer,
>
> We appreciate your insightful comments and provide our response as follows.
>
> 1. **I curious about if the method is scalable to high dimension data.**
>
>    **Response**:   Thank you for the insightful comments. In this paper we primarily test our method on multi-dimensional time-series datasets, leveraging temporal locallity. Despite the resource intensities and robustness requirements of explaining higher-dimensional data, we believe our method can be scaled if additional modality-specific adaptations, beyond temporal correlation, are made. Taking video sequences as an example, perturbations and corresponding salience learning can be implemented on the spatial and temporal streams separately [1, 2]. Our method can be extended with a similar philosophy of handling time and vision features orthogonally, e.g., by adding another trend function to the spatial dimension. This presents an intriguing avenue for future research.
>
> 2. **What is the and values you use for each dataset? and how do you determine their values?**
>
>    **Response**: Thanks for the nice question. In fact, we have given the specific parameters chosen for each dataset, see Table 6 in Appendix D.4. For your convenience, we list them as follows:
>
>    | Parameter         | Rate-Time      | Rate-Observation        |     Switch-Feature     |     State      |     MIMIC-III      |
>    | :-------------: | :-----------------: | :-----------------: | :---------------: | :---------------: | :---------------: |
>    | $\alpha$ | 0.1       | 0.1              |   1.0   |  2.0  |   0.005   |
>    | $\beta$    | 0.1       | 0.1              |   2.0   |  1.0  |   0.01   |
>
>    Specifically, the four Rate experiments used a default value of 0.1 determined by experience and other mask-based methods. On this basis, we perform ablation experiments (See Tables 9 and 10 in Appendix E.2) around the default values and find the best parameters applicable to the Switch-Feature and State dataset. Note that since the proportion of salient features in the data is unknown, finding a subset of features is unstable for most of the interpretation methods by picking the regularization hyperparameter. We mentioned that it's in the limit: "However, an inherent limitation of our method is the selection of sparse parameters, especially when dealing with different datasets..."
>
>    For the MIMIC-III dataset, we add the same ablation experiments in the latest version. We employ Accuracy and Cross-Entropy as metrics and show the average substitution in the following table. This Table shows that $\beta$ is best set to $0.01$ to learn counterfactual perturbations. Notably, lower values of $\alpha$ yield superior results, but excessively regularizing $m$ toward $0$ may prove disadvantageous [3]. Therefore, we select $\alpha=0.005$ and $\beta=0.01$ as deterministic parameters on the MIMIC-III dataset.
>
>
> Table 11: Effects of $\alpha$ and $\beta$  on MIMIC-III mortality. We mask 20\% data and replace the masked data with the overall average over time for each feature. **bold** is the best and values are reported as Accuracy - Cross-Entropy.
>
> |                |              $\alpha$= 0.001              |              $\alpha$= 0.005              |               $\alpha$= 0.01               |               $\alpha$= 0.1                |               $\alpha$= 1.0               |
> | :------------: | :---------------------------------------: | :---------------------------------------: | :----------------------------------------: | :----------------------------------------: | :---------------------------------------: |
> | $\beta$= 0.001 | ${0.982}{\pm0.003}$ - ${0.124}{\pm0.007}$ | ${0.983}{\pm0.003}$ - ${0.122}{\pm0.007}$ | ${0.984}{\pm0.002}$ - ${0.120}{\pm0.006}$  | ${0.993}{\pm0.001}$ - ${0.094}{\pm0.004}$  | ${0.997}{\pm0.001}$ - ${0.087}{\pm0.004}$ |
> | $\beta$= 0.005 | ${0.981}{\pm0.002}$ - ${0.123}{\pm0.007}$ | ${0.984}{\pm0.002}$ - ${0.123}{\pm0.006}$ | ${0.984}{\pm0.003}$ - ${0.121}{\pm0.007}$  | ${0.993}{\pm0.002}$ - ${0.095}{\pm0.006}$  | ${0.996}{\pm0.001}$ - ${0.087}{\pm0.005}$ |
> | $\beta$= 0.01  | ${0.980}{\pm0.003}$ - ${0.124}{\pm0.007}$ |  **0.980$\pm$0.002 -  0.127$\pm$0.007**   | ${0.984}{\pm0.002}$ - ${0.121}{\pm0.007}$  | ${0.994}{\pm0.002}$ - ${0.094}{\pm0.004}$  | ${0.996}{\pm0.001}$ - ${0.087}{\pm0.004}$ |
> |  $\beta$= 0.1  | ${0.980}{\pm0.003}$ - ${0.127}{\pm0.007}$ | ${0.980}{\pm0.003}$ - ${0.127}{\pm0.007}$ | ${0.983}{\pm 0.003}$ - ${0.123}{\pm0.007}$ | ${0.992}{\pm0.002}$ - ${0.098}{\pm 0.006}$ | ${0.997}{\pm0.001}$ - ${0.087}{\pm0.005}$ |
> |  $\beta$= 1.0  | ${0.981}{\pm0.002}$ - ${0.127}{\pm0.006}$ | ${0.981}{\pm0.003}$ - ${0.128}{\pm0.008}$ | ${0.983}{\pm0.002}$ - ${0.123}{\pm0.007}$  | ${0.989}{\pm0.002}$ - ${0.106}{\pm0.007}$  | ${0.996}{\pm0.001}$ - ${0.088}{\pm0.005}$ |

---

> ### Author Response · Authors · 2023-11-19
> **Response to Reviewer zXTo (part 2/2)**
>
> 3. **Contrastive learning about distance choices in section 4.1.**
>
>    **Response**: We thank the reviewer for the insightful comment. Most explanation methods, such as Dynamask and Extrmask, construct the loss (mask or perturbation) using the $l_1$ norm. In Eq.(2), the first term encourages the original sample $x_i$ to differ from the perturbed sample $x^r_i$. The second term, akin to a regularization enforcing Manhattan distance from $0$ as $|x^r_i - 0|$, constrains the range of counterfactuals. We therefore intuitively used the Manhattan distance measure of similarity between samples.
>
>    To better relieve your concern, we add the effect of distance type in contrastive learning in the latest version (see Table 8 in Appendix E.1). We evaluate three typical distance metrics in Rare-Time and Rare-Observation datasets: Manhattan distance, Euclidean distance, and cosine similarity, and we list it as follows. The results indicate that the Manhattan distance is slightly better than the other examined losses.
>
>
> Table 8: Performance of ContraLSP with different contrastive loss types on rare experiments.
> | $\mathcal{L}_{cntr}$ | AUP    |        AUR        |     $I_m$      |     $S_m$     |
> | :-------------: | :-----------------: | :-----------------: | :---------------: | :---------------: |
> |  |  | Rare-Time |                |  |
> | Manhattan distance | **1.00**$\pm$ **0.00** | **0.97**$\pm$ **0.01** |   19.51$\pm$ 0.30   |  **4.65** $\pm$ **0.71**  |
> | Euclidean distance | **1.00**$\pm$ **0.00** |     0.97$\pm$ 0.02     | **19.67**$\pm$ **0.52** |     4.97 $\pm$ 0.55     |
> | cosine similarity | **1.00**$\pm$ **0.00** | 0.96$\pm$ 0.02 | 18.41$\pm$ 0.64 | **5.87**$\pm$ **0.74** |
> |  |  | Rare-Observation |  |  |
> | Manhattan distance | **1.00**$\pm$ **0.00** | **1.00**$\pm$ **0.00** | 20.68$\pm$ 0.03 | **0.32** $\pm$ **0.16** |
> | Euclidean distance | **1.00**$\pm$ **0.00** | 1.00$\pm$ 0.01 | **20.72**$\pm$ **0.06** | 0.69 $\pm$ 0.17 |
> | cosine similarity | **1.00**$\pm$ **0.00** | 0.98$\pm$ 0.01 | 19.22$\pm$ 0.06 | 0.98 $\pm$ 0.23 |
>
> [1] Kowal, et al. A Deeper Dive Into What Deep Spatiotemporal Networks Encode: Quantifying Static vs. Dynamic Information. In *CVPR*, 2022.
>
> [2] Li, et al. Towards visually explaining video understanding networks with perturbation. In *WACV*, 2021.
>
> [3] Joseph Enguehard. Learning perturbations to explain time series predictions. In *ICML*, 2023.

---

> > ### Author Response · Authors · 2023-11-21
> > **Thank you for the review! Have we clearly addressed the concerns?**
> >
> > We greatly appreciate the time you took to review our paper. Due to the short duration of the author-reviewer discussion phase, we would appreciate your feedback on whether your main concerns have been adequately addressed.
> >
> > We are ready and willing to provide further explanations and clarifications if necessary.
> >
> > Thank you very much!

---

> > > ### Comment · Reviewer_zXTo · 2023-11-23
> > > **Thank you**
> > >
> > > Thank you for your response. However, as this paper does not fully match my expertise of research, I would keep my score and let the AC and other reviewers to make the decision

---

### Official Review · Reviewer_mcnc · 2023-11-02

**Soundness:** 3 good
**Presentation:** 3 good
**Contribution:** 2 fair
**Rating:** 6
**Confidence:** 3

**Summary:**

The paper presents a tool for time series explanations, which is challenging as it requires matching complex temporal patterns and features. Perturbations have been among popular approaches to identify counterfactuals, but the paper argues, in time series these can be particularly challenging since perturbations can make samples OOD, rendering the resulting explanation meaningless to the original goal (i.e., finding a counterfactual). This is especially the case when considering the label-free perturbation scenario, which is less studied in literature.

In order to address this, the paper proposes a label-free Contrastive, and locally sparse perturbation approach that is more likely to generate in-domain perturbations.

ContraLSP has two main components -- first a contrasting objective that seeks to ensure perturbations are dissimilar from the the original time series and are "more distracting". There is also a sparse, stochastic gate for each feature to ensure sparsity in feature selection. The final objective contains 3 terms -- contrasting loss, a regularizer on the mask, and a proximal loss to ensure predictions are close to the original, un perturbed input.

**Strengths:**

* **Problem statement and motivation**: Time series explainability is an important topic, that has received relatively lesser attention. The paper correctly identifies the trouble with OOD perturbations in time varying data, which are poorly understood in comparison with image and language modalities. The tool is capable of working with both blackbox and whitebox models, as well as working with regression and classification tasks-- which are positives.
* **Problem formulation**: The contrasting approach to time series explainability appears to be novel as far as I know.
* **Evaluation**: Extensive empirical evaluations are conducted on synthetic benchmarks with available ground truth on feature importance, as well as real world clinical data. ContraLSP appears to be outperforming several related baselines in both scenarios.

**Weaknesses:**

* **Counterfactuals**: The paper generously uses counterfactuals in the text to indicate perturbations produced by their model, whereas this is a label free approach and the mask is learned to minimize the gap between the original and unperturbed samples. The contrasting objective is the only source for potential counterfactual generation, that too it is not guaranteed to do so --  this distinction should be made more explicit in the text, and reduce the usage of perturbations being called counterfactuals. The experiments mostly only measure the ability of ContraLSP on identifying salient features, so this claim should be tempered down.
* In this context, can the authors elaborate on the failures or weaknesses of ContraLSP? Specifically, when is it expected to fail, perhaps in comparison to techniques that work with labeled data?
* **The contrasting objective** :  Since negatives are chosen at random, they are likely going to be weak negatives, and claiming these will be "more counterfactual" is probably not true. It must also be defined what "more counterfactual" means here -- more than what? How does the random negative selection ensure perturbations are crossing over class boundaries? Why is an L1 edit distance the right distance metric to do this?
* **OOD Perturbations**  how does this objective _guarantee_ or at least ensure lack of OOD perturbations? Is the training of the perturbation function with contrastive loss sufficient to ensure this?
* **Sparse gating and stochasticity**: Please define what the heavy-tailed nature of the sparse feature selection is, and how it is relevant to ContraLSP. The hard thresholding function in eqn (3) is only needed due to the random noise injected in the masking, since $\mu'$ is a sigmoid function already.. why is the noise needed in the first place?
* The paper's writing is not easy to follow, and this makes it hard to assess the core contribution of the work more rigorously. There are a lot of vague statements which are not stated clearly. Some of these are listed below:
	* ".. perturbation may not alleviate the distribution shift issue.." (in the abstract)
	* ".. unregulated data distribution .." (in Sec 3)
	* ".. allows perturbed features to perceive heterogenous samples, thus increasing the impact of the perturbation.." (Sec 4)
	* ".. counterfactual perturbations more distracting.." (Sec 4.1)
	* ".. due to their heavy tailed nature.." (Sec 4.2)
* **Illustrations**: I recognize time series explanations are more challenging than visual data like imagery. However, the current set of illustrations are not very clear. For e.g. in Fig 5 why is the sum of salient observations shown? What is the inference from this figure? its very unclear, please make the key observations more explicit, perhaps with the help of a simpler dataset or time series and more consistent with Fig 1, which is easier to follow.

**Questions:**

Please see above, I have listed several questions.

---

> ### Author Response · Authors · 2023-11-19
> **Response to Reviewer mcnc (part 1/3)**
>
> Dear Reviewer,
>
> Thank you for your careful reading and insightful suggestions. We answer the questions below:
>
> 1. **Counterfactuals:  The contrasting objective might not guarantee counterfactual generation. In this context, can the authors elaborate on the failures or weaknesses of ContraLSP with labeled data?**
>
>    **Response**: Thanks for pointing this question out. Regarding the use of counterfactuals in our paper, we acknowledge the need to clarify the distinction between our label-free perturbation and traditional counterfactual perturbation [1]. Traditional explanations select counterfactuals through different labels. In contrast, our model utilizes contrastive learning to minimize the difference between perturbed and negative samples, primarily driven by the distance to the current sample. We need to consider regression and classification tasks where the selection of counterfactuals is not satisfied by labeling, as using labeled data conflicts with explaining black-box models. Instead, we believe that the generation of negative samples can be tended to counterfactuals through unsupervised methods [2], such as top-k prediction and clustering. However, this is beyond the scope of this work. We have added it to the limitation in the new version and left it for future work.
>
>
> [1] Goyal, et al. Counterfactual visual explanations. ICML, 2019.
>
> [2] UNR-Explainer: Counterfactual Explanations for Unsupervised Node Representation Learning Models. 2023.
>
> 2. **Define and clarify what "more counterfactual" means. How does the random negative selection ensure perturbations are crossing over class boundaries?**
>
>    **Response**:  Thanks for your constructive comment. We apologize for the "more counterfactual" that is incorrect and have modified it in the latest version. We believe the selection of negative samples by random, rather than by the furthest distance because we need to consider the classification task.  By intuitively opting for random selection, we aim to diversify the negative samples across different categories, rather than solely focusing on those distantly positioned from the current category. Previous studies [3, 4] have also depended on the quality of randomly-selected negative samples. Of course, these samples often lack complexity, making them easily distinguishable and offering limited supervised signals [5]. How to select negative samples more rationally for perturbation remains an area requiring exploration, which we intend to address in future research endeavors.
>
> [3] Huang, et al. Cross-sentence temporal and semantic relations in video activity localisation. In *ICCV*, 2021.
>
> [4] Zhang, et al. Counterfactual contrastive learning for weakly-supervised vision-language grounding. In *NeurIPS*, 2020.
>
> [5] Zheng M, et al. Weakly supervised temporal sentence grounding with gaussian-based contrastive proposal learning. In *CVPR*, 2022.
>
> 3. **Why is an L1 edit distance the right distance metric to do this?**
>
>    **Response**:  Thank you for the thoughtful comment. Most explanation methods, like Dynamask and Extrmask, utilize the $l_1$ norm to construct the loss (mask or perturbation). In Eq. (2), the first term encourages the original sample $x_i$ to differ from the perturbed sample $x^r_i$. The second term, akin to a regularization enforcing Manhattan distance from $0$ as $|x^r_i - 0|$, constrains the range of counterfactuals. We therefore intuitively used the Manhattan distance measure of similarity between samples.
>
>    In response to concerns, we incorporate the effect of distance type in contrastive learning in the latest version (see Table 8 in Appendix E.1). We evaluate three typical distance metrics in Rare-Time and Rare-Observation datasets: Manhattan distance, Euclidean distance, and cosine similarity, and we list it as follows. Our results indicate that the Manhattan distance exhibits slightly superior performance compared to the other evaluated metrics.
>
> Table 8: Performance of ContraLSP with different contrastive loss types on rare experiments.
> | $\mathcal{L}_{cntr}$ | AUP    |        AUR        |     $I_m$      |     $S_m$     |
> | :-------------: | :-----------------: | :-----------------: | :---------------: | :---------------: |
> |  |  | Rare-Time |                |  |
> | Manhattan distance | **1.00**$\pm$ **0.00** | **0.97**$\pm$ **0.01** |   19.51$\pm$ 0.30   |  **4.65** $\pm$ **0.71**  |
> | Euclidean distance | **1.00**$\pm$ **0.00** |     0.97$\pm$ 0.02     | **19.67**$\pm$ **0.52** |     4.97 $\pm$ 0.55     |
> | cosine similarity | **1.00**$\pm$ **0.00** | 0.96$\pm$ 0.02 | 18.41$\pm$ 0.64 | **5.87**$\pm$ **0.74** |
> |  |  | Rare-Observation |  |  |
> | Manhattan distance | **1.00**$\pm$ **0.00** | **1.00**$\pm$ **0.00** | 20.68$\pm$ 0.03 | **0.32** $\pm$ **0.16** |
> | Euclidean distance | **1.00**$\pm$ **0.00** | 1.00$\pm$ 0.01 | **20.72**$\pm$ **0.06** | 0.69 $\pm$ 0.17 |
> | cosine similarity | **1.00**$\pm$ **0.00** | 0.98$\pm$ 0.01 | 19.22$\pm$ 0.06 | 0.98 $\pm$ 0.23 |

---

> ### Author Response · Authors · 2023-11-19
> **Response to Reviewer mcnc (part 2/3)**
>
> 4. **OOD Perturbations how does this objective guarantee or at least ensure lack of OOD perturbations? Is the training of the perturbation function with contrastive loss sufficient to ensure this?**
>
>    **Response**:  Thank you for the thoughtful comment! To investigate whether the perturbed samples are within the original dataset's distribution, we add the analysis experiments of perturbation distribution in the latest version (see Appendix F). We first compute the distribution of the original samples by kernel density estimation (KDE). Subsequently, we assess the log-likelihood of each perturbed sample under the original distribution, called as KDE-score, where closer to $0$ indicates a higher likelihood of perturbed samples originating from the original distribution. Additionally, we quantify the KL-divergence between the distribution of perturbed samples and original samples, where a smaller KL means that the two distributions are closer. We conduct experiments on the Rare-Time and Rare-Observation datasets and the results are shown in Table 12. It demonstrates that our ContraLSP's perturbation is more akin to the original distribution compared to the zero and mean perturbation. Furthermore, Extrmask performs best because it generates perturbations only from current samples, and therefore the generated perturbations are not guaranteed to be uninformative. This conclusion aligns with the visualization depicted in Figure 1. If you have any additional questions or comments, we would be happy to have further discussions.
>
> Table 12: Difference between the distribution of different perturbations and the original distribution.
> |  | Rare-Time |        Rare-Time        |    Rare-Observation     |    Rare-Observation     |
> | :-------------: | :-----------------: | :-----------------: | :---------------: | :---------------: |
> | Perturbation type | KDE-score$\uparrow$  |     KL-divergence   $\downarrow$  | KDE-score $\uparrow$ | KL-divergence $\downarrow$ |
> | Zero perturbation | -25.242 | 0.0523 | -23.377 |  0.0421|
> | Mean perturbation | -30.805 |  0.0731 | -26.421 |  0.0589|
> | Extrmask perturbation |-22.532 |  0.0219 | -19.102 | 0.0104 |
> | ContraLSP perturbation |  -23.290| 0.0393 |  -22.732 | 0.0386|
>
>
>
> 5. **Please define what the heavy-tailed nature of the sparse feature selection is, and how it is relevant to ContraLSP.**
>
>    **Response**: Thanks for your constructive suggestion. The process of feature selection and explanation necessitates stability in the chosen feature set. However, employing logistic masks introduces notable variance in the approximated Bernoulli variables due to their heavy-tailedness [6], which like Figure 1 Top left in [6], often resulting in inconsistencies within the selected feature set. Therefore, we utilize the Bernoulli continuous relaxation method based on Gaussian variables, represented as the $\ell_0$-like constraint in Eq. (5). ContraLSP incorporates sample-specific sparse gates as mask indicators that are optimized by the $\ell_0$-like constraint, producing binary-skewed masks specific to time series data. In addition, we ensure the consistency of the underlying time series patterns by considering temporal trends.
>
>
> [6] Yamada, et al. Feature selection using stochastic gates. In *ICML*, 2020.
>
>
> 6. **About the random noise injected in Eq (3) and the sigmoid-weighted unit to smooth $\mu^\prime$**
>
>    **Response**: Thanks for the nice question, but it's not right. To prevent temporal abruptness and jump saliency, we adopt the sigmoid-weighted unit that $\mu^\prime=\mu*\sigma(\tau*\mu)$ to smooth the mask. Thus, $\mu^\prime\in \mathbb{R}$ where is learnable, it is just smoothed by the sigmoid function. The reason why we inject a Gaussian noise on $mu^\prime$ is that the $\ell_0$-like constraint is used instead of the traditional $\ell_1$ regularization in Eq. (5), which makes the mask more binary-skewed [6] (i.e., biased towards 0 or 1 but smoothness).

---

> ### Author Response · Authors · 2023-11-19
> **Response to Reviewer mcnc (part 3/3)**
>
> 7. **The paper's writing is not easy to follow, and this makes it hard to assess the core contribution of the work more rigorously...**
>
>    **Response**: Thanks for pointing this question out. We appreciate the concerns raised regarding the clarity and motivation within our manuscript. We are committed to revising the document to improve its organization, ensuring a coherent introduction of terms for enhanced comprehension. The latest version has corrected inappropriate sentences, including some of the issues you mentioned, and a comprehensive review will be conducted for the final version.
>
>    Our contribution entails ContraLSP, a stronger explanatory tool for time series models. It utilizes counterfactual samples that have a large distance from the current sample to construct uninformative perturbations through contrastive learning. ContraLSP also integrates sample-specific sparse gates, creating binary-skewed masks on time series, and incorporates temporal trends for consistent alignment of latent patterns. We validate our method via experiments on synthetic and real-world time series datasets, including classification and regression tasks.  It allows quantitative comparisons with state-of-the-art time series explainers.
>
>
> 8. **Improve figure clarity for better reader understanding, especially in Fig. 5.**
>
>
>    **Response**: We appreciate the reviewer's feedback regarding the clarity of our illustrations, particularly Figure 5. We present a comparison between the perturbations generated by ContraLSP and Extrmask on the Rare-Observation (Diffgroups) experiment, as shown in Figure 5 (the updated version). We randomly select a sample in each of the two groups and sum all observations. ContraLSP provides counterfactual information, yet Extrmask's perturbation is close to 0. This suggests ContraLSP provides counterfactual perturbations that are more beneficial to non-salient areas while emphasizing the mask more distinctly in salient areas compared to Extrmask. This is consistent with the findings in Figure 1. In addition, we show the perturbation of summed observations, leveraging statistical features to understand this dataset. This is because there are 50 observations in this experiment, but only 5 salient observations are included.
>
> We thank you again for your constructive comments and for your efforts to improve the quality of our paper.

---

> > ### Comment · Reviewer_mcnc · 2023-11-21
> >
> > Thank you for the detailed responses, you have answered most of my concerns. I will raise my score to 6.

---

### Author Response · Authors · 2023-11-19
**General Response**

Dear AC and Reviewers,

We would like to sincerely thank the reviewers for their positive feedback and highly constructive suggestions. For clarity and readability of the paper, the following changes have been made to the manuscript and a new version uploaded.

* We have refined the paragraphs in Section 5.1 to make the white-box regression experiment more readable.
* We improved Figure 5 to make it more understandable.
* We added limitations and future work to our conclusion section.
* We added the effect of distance type and hyperparameter selection described in Appendix E.
* We added a distribution analysis of perturbations in Appendix F.
* Minor change of wordings and re-description for conciseness.

Thanks again,

The Authors

---

### Meta-Review · Area_Chair_sMKr · 2023-12-08

**Metareview:**

The proposed paper deals with time series explanability and introduces a model which uses counterfactual samples to build uninformative perturbations but keeps distribution using contrastive learning. It has received 4 reviews, which were initially leaning towards rejection, mixed. Reviewers appreciated

- A novel approach,
- the usage of contrastive learning for this task,
- a well-written paper,
- robustness of the model,
- Extensive evaluations and comparisons,
- performance improvements.

The reviewers also raised weaknesses and issues, in particular on the role of the counterfactuals, details of the contrastive formulation, the role of OOD perturbations, sensitivity to hyper-parameters, some aspects of the experimental setup linked to comparisons and metrics.

The authors provided answers which convinced several reviewers and lead to raised scores, some of them substentially so. Reviewer 9mjj raised their score from 3 to 5 but remained sceptical on some aspects, providing the only (slightly) negative rating. The remaining doubts were on the significance of the results.

The AC weighted the rich input from the discussion and judged that the paper presents a sufficient interest to the field and recommends acceptance.

**Justification For Why Not Higher Score:**

The paper was borderline, there is no reason to increase.

**Justification For Why Not Lower Score:**

This paper could have been rejected, but a majority of the reviewers recommended acceptance
The remaining slightly reserved reviewer was a bit contradictory in their assessment, in particular on novelty.

---

### Decision · Program_Chairs · 2024-01-16

Accept (poster)